# A novel circulating tamiami mammarenavirus shows potential for zoonotic spillover

Hector Moreno[1]*, Alberto Rastrojo[2,3], Rhys Pryce[4], Chiara Fedeli[1], Gert Zimmer[5,6], Thomas A. Bowden[4], Gisa Gerold[7,8,9], Stefan Kunz[1†]

**1** Institute of Microbiology, Lausanne University Hospital (IMUL-CHUV), Lausanne, Switzerland, **2** Department of Virology and Microbiology, Centro de Biología Molecular Severo Ochoa (CBMSO-CSIC), Madrid, Spain, **3** Genetic Unit, Department of Biology, Universidad Autónoma de Madrid, Madrid, Spain, **4** Division of Structural Biology, Wellcome Centre for Human Genetics, University of Oxford, Roosevelt Drive, Oxford, United Kingdom, **5** Institute of Virology and Immunology (IVI), Mittelhäusern, Switzerland, **6** Department of Infectious Diseases and Pathobiology (DIP), Vetsuisse Faculty, University of Bern, Bern, Switzerland, **7** TWINCORE -Center for Experimental and Clinical Infection Research, Institute for Experimental Virology, Hannover, Germany, **8** Department of Clinical Microbiology, Virology & Wallenberg Centre for Molecular Medicine (WCMM), Umeå University, Umeå, Sweden, **9** Department of Biochemistry, University of Veterinary Medicine Hannover, Hannover Germany

† Deceased on 10/1/2020
* hector.moreno@chuv.ch

**Data Availability Statement:** All relevant data are within the manuscript and its Supporting information files. All NGS files are available from

## Abstract

A detailed understanding of the mechanisms underlying the capacity of a virus to break the species barrier is crucial for pathogen surveillance and control. New World (NW) mammarenaviruses constitute a diverse group of rodent-borne pathogens that includes several causative agents of severe viral hemorrhagic fever in humans. The ability of the NW mammarenaviral attachment glycoprotein (GP) to utilize human transferrin receptor 1 (hTfR1) as a primary entry receptor plays a key role in dictating zoonotic potential. The recent isolation of Tacaribe and lymphocytic choriominingitis mammarenaviruses from host-seeking ticks provided evidence for the presence of mammarenaviruses in arthropods, which are established vectors for numerous other viral pathogens. Here, using next generation sequencing to search for other mammarenaviruses in ticks, we identified a novel replication-competent strain of the NW mammarenavirus Tamiami (TAMV-FL), which we found capable of utilizing hTfR1 to enter mammalian cells. During isolation through serial passaging in mammalian immunocompetent cells, the quasispecies of TAMV-FL acquired and enriched mutations leading to the amino acid changes N151K and D156N, within GP. Cell entry studies revealed that both substitutions, N151K and D156N, increased dependence of the virus on hTfR1 and binding to heparan sulfate proteoglycans. Moreover, we show that the substituted residues likely map to the sterically constrained trimeric axis of GP, and facilitate viral fusion at a lower pH, resulting in viral egress from later endosomal compartments. In summary, we identify and characterize a naturally occurring TAMV strain (TAMV-FL) within ticks that is able to utilize hTfR1. The TAMV-FL significantly diverged from previous TAMV isolates, demonstrating that TAMV quasispecies exhibit striking genetic plasticity that may facilitate zoonotic spillover and rapid adaptation to new hosts.

the European Nucleotide Archive (ENA) database (accession number PRJEB31100).

**Funding:** This research was supported by Swiss National Science Foundation grants SINERGIA Nr. CRSII3_160780/1, 310030_170108 to S.K. and funds to S.K. from the University of Lausanne. G.G. was supported by the Knut and Alice Wallenberg Foundation. T.A.B. is supported by the Medical Research Council (MR/S007555/1). The Wellcome Centre for Human Genetics is supported by Wellcome Centre grant 203141/Z/16/Z (T.A.B.). The funders had no role in study design, data collection and analysis, decision to publish, or preparation of the manuscript.

**Competing interests:** The authors have declared that no competing interests exist. Author Prof. Stefan Kunz was unable to confirm their authorship contributions. On their behalf, the corresponding author has reported their contributions to the best of their knowledge.

## Author summary

Mammarenaviruses include emergent pathogens responsible of severe disease in humans in zoonotic events. The ability to use the human Transferrin receptor 1 (hTfR1) strongly correlates with their pathogenicity in humans. We isolated a new infectious Tamiami virus strain (TAMV-FL) from host-seeking ticks, which, contrary to the previous rodent-derived reference strain, can use hTfR1 to enter human cells. Moreover, serial passaging of TAMV-FL in human immunocompetent cells selected for two substitutions in the viral envelope glycoprotein: N151K and D156N. These substitutions increase the ability to highjack hTfR1 and the binding capacity to heparan sulfate proteoglycans and cause delayed endosomal escape. Our findings provide insight into the acquisition of novel traits by currently circulating TAMV that increase its potential to trespass the inter-species barrier.

## Introduction

Mammarenaviruses comprise a large and diverse group that includes several emerging zoonotic viruses that cause severe hemorrhagic fever (HF) disease in humans [1]. Available treatment options for mammarenavirus infection include administration of convalescent plasma and the off label use of ribavirin, which has limited antiviral efficacy and is frequently associated with significant side effects [2–4]. Mammarenaviruses are enveloped viruses with a negative-strand RNA genome, which replicate in the cytoplasm of the host cell. The viral genome comprises two RNA segments, a small (S) segment that encodes the envelope glycoprotein precursor (GPC) and the nucleoprotein (NP), and a large (L) segment that encodes the matrix protein (Z) and the viral RNA-dependent RNA polymerase (L). The GPC precursor is cleaved by cellular proteases to form the stable signal peptide (SSP) and the mature surface glycoproteins GP1 and GP2 [5]. Mature spikes, termed GP, are composed of trimers of GP1/GP2/SSP subunits that form the functional unit for virus attachment and entry [5–8]. GP is a key determinant of species and cell tropism and represents the major target for neutralizing antibodies [9–11].

Mammarenaviruses are divided into Old World (OW) and New World (NW) groups based on their antigenic properties, phylogeny, and geographic distribution [12]. The phylogenetically diverse NW mammarenaviruses are subdivided into the clades A, B, and C, which are found in South America, and clade D (formerly A/B) which is restricted to North America [13]. NW arenaviruses causing hemorrhagic fevers belong to clade B and include Junín (JUNV), Machupo (MACV), Guanarito (GTOV), Sabia (SABV), and Chapare virus (CHAV). Within clade B, pathogenic viruses do not form a defined subgroup, but cluster into sub-lineages along with non-pathogenic viruses. Whilst the GPs of clade D NW arenaviruses are closely related to those in clade B, the NPs are more related to those in clade A, probably as the result of recombination between clades A and B [14]. The principal reservoir hosts for NW mammarenaviruses are rodents of the subfamilies *Neotominae* and *Sigmodontinae* of the *Cricetidae* family, with the exception of Tacaribe virus (TCRV), which has been isolated from fruit bats [15]. Despite sporadic nosocomial human-to-human infections, highly pathogenic mammarenavirus infections occur mainly by zoonotic transmission through inhalation of aerosolized contaminated material and can result in severe hemorrhagic fever disease with case-fatality rates of 15–30% [16–19]. The identification of WWAV AV96010151 (WWAV-AV96)

strain associated with a small number of human fatalities suggested potential for viral emergence among clade D NW mammarenaviruses [20].

A detailed understanding of the mechanisms that dictate cross-species virus transmission is crucial for surveillance and the evaluation of the potential for virus emergence. Since receptor usage is known to be a key determinant of cross-species transmission and human disease potential amongst NW mammarenaviruses, an understanding of the factors that dictate receptor utilization and adaptation is crucial for assessing zoonotic potential [12,21–28]. Clades B and D NW mammarenaviruses use the conserved cargo-receptor transferrin receptor 1 (TfR1) for cell entry, and their zoonotic potential is linked to the ability to utilize the human orthologue (hTfR1) [21,22,25,27–29]. Given that relatively minor changes to NW GP1s can facilitate hTfR1 recognition and that mammarenavirus are subjected to quasispecies dynamics, which allow rapid adaptation to environmental change [14,30,31], clade D mammarenaviruses represents an important reservoir with the potential for emergence as human pathogens [21,23,27].

Heparan sulfate proteoglycans (HSPG) are abundant in airway epithelial cells [32–34] and can act as attachment factors for several viruses whereby they increase local virion concentration at the cellular membrane. Although HSPG binding has been studied in several viruses [35–40], its importance for NW mammarenaviruses is currently unknown. Binding to HSPG depends on electrostatic interactions occurring between the negatively charged polysaccharide component of the membrane and regions of positive charged on viral surface proteins. Given the importance of HSPG as an attachment factor for many viruses, HSPG binding may be a relevant factor during viral adaptation to new cell types and hosts.

Whilst NW arenavirus spillover events typically occur in restricted geographic foci associated with the home range of host rodents, the involvement of arthropod vectors may augment transmission dynamics, increasing the likelihood of zoonotic transmission. Although further investigations are still needed to clarify the biological role of arthropods as vectors in arenavirus transmission, the recent isolation of TCRV from the host-seeking tick *Amblyomma americanum* in Florida and the genomic characterization of new lymphocytic choriominingitis virus (LCMV) strains in several tick species in China, already provided the first evidence of the presence of mammarenaviruses in arthropods [41,42]. We screened *A. americanum* for other arenaviruses and identified a novel tick-derived variant of Tamiami virus (TAMV), referred to as TAMV-Florida (FL), which unlike previously isolated TAMV strain utilizes hTfR1 to infect human cells, and whose nucleotide and amino acid sequences are divergent from previously reported TAMV sequences.

The unique non-reciprocal superinfection exclusion capacity of TAMV [43] allowed us to use serial passaging on immunocompetent susceptible human cells (A549) for isolating TAMV-FL, which was largely outcompeted by TCRV in the original tick-derived sample. During serial passaging, the quasispecies of TAMV-FL acquired and rapidly enriched the mutations causing the amino acid substitutions N151K and D156N in GP. Characterization of these mutations allowed us to observe that these changes affect key factors involved in the infection process, such as hTfR1 usage, affinity to HSPG and pH threshold for triggering membrane fusion. In sum, this study provides the first evidence of a replication competent clade D NW mammarenavirus, TAMV, in arthropods. Furthermore, we show that TAMV-FL can utilize hTfR1 for cellular entry, and has the adaptive capacity to rapidly adjust to cells of new host species. Our findings uncover the potential for zoonosis of TAMV-FL and highlight the importance of functional evaluation of viruses circulating in nature in order to assess potential risks of spillover.

## Results

### Detection of a novel TAMV variant in host-seeking *Amblyomma americanum* ticks

The advent of novel powerful NGS approaches sharply increased the discovery rates of new emerging viruses [44]. We used an unbiased NGS approach to screen samples of *A. americanum* ticks trapped in Florida [41], which were previously found positive for TCRV, for the presence of additional mammarenavirus sequences. To this end, 100 ticks were pooled to prepare cleared homogenates [45] and subjected to three blind passages on VeroE6 cells [41]. We used tissue culture supernatants (TCS) of the third passage to prepare cDNA libraries for Illumina-based NGS analysis. To facilitate detection of under-represented viral sequences, we employed a step-wise strategy (Fig 1A). In our NGS run, we found 99158 and 621 out of 105780 total filtered reads corresponding to RNA of TCRV and TAMV, respectively (S1 Table). For further analysis, candidate arenavirus sequence reads were assembled in contigs longer than 500 bp and subjected to BLAST analysis against available reference viral genomes (Fig 1A and S2 Table). Obtained reads were assembled into 4 contigs corresponding to the complete L and S segments of TCRV and TAMV (S2 Table). Comparison between the obtained TCRV genomic nucleotide sequences (TCRV-FL. Genbank accession numbers MW150032.1 (L) and MW150033.1 (S), respectively) with the TCRV reference strain 11573 (MT478050.1 and MT478051.1) [46], reveals overall identities of 99.87% and 99.91% in segments L and S, respectively (Fig 1B and 1E, and S2 and S3 Tables), confirming the remarkably high similarity between tick-derived TCRV-FL and the TCRV reference strain. Nevertheless, despite the similar origin of the TCRV-FL and the TCRV Florida isolate [41], the comparison with other available TCRV sequences revealed TCRV-BEI strain [46] as the closest relative of TCRV-FL (S3 Table). Alignment of the obtained reads against the rodent-borne TAMV reference strain W-10777 (TAMV-Ref. NC_010701.1 and NC_010702.1) [47] reveals a high sequence variation despite uneven and limited coverage (Fig 1C). Therefore, we next assembled a full-length tick-derived TAMV genome (TAMV-FL. GenBank access numbers MK500936 and MK500937) from the reads obtained in our NGS run (Fig 1D). In contrast to the tick-derived TCRV, the nucleotide and amino acid sequences of the assembled tick-derived TAMV-FL differ significantly from the TAMV-Ref isolated from the cotton rat *Sigmodon hispidus* in 1965, in Florida, and other available TAMV sequences [48] (Table 1 and Fig 1E and S1 Fig). The complete sequences of the S and L segments of TAMV-FL contained 3526 and 7143 nucleotides, respectively (S2 Table). The S segment shows 85.28% identity at the nucleotide level and 89.93% and 90.33% at the amino acid level of NP and GPC, respectively, compared to TAMV-Ref (Fig 1E and Table 1). The identity in the L segment at the nucleotide level is 87.18% and at the amino acid level 95.29% and 91.77% for the Z protein and the viral polymerase L, respectively (Fig 1E). The extent of sequence variation is incompatible with a laboratory contamination and raised the possibility of TAMV-FL being a new mammarenavirus species. The current species definition of mammarenaviruses is based on several criteria, including association with a specific host species, geographic distribution, status as human disease agent, lack of serological cross-neutralization, and the results of pairwise sequence comparisons (PASC) of coding-complete genomes using >12% divergence in NP amino acid sequence as one of the criteria for species demarcation [12]. Despite significant sequence deviation, TAMV-FL fulfills the criteria to be considered a strain of the same species as TAMV.

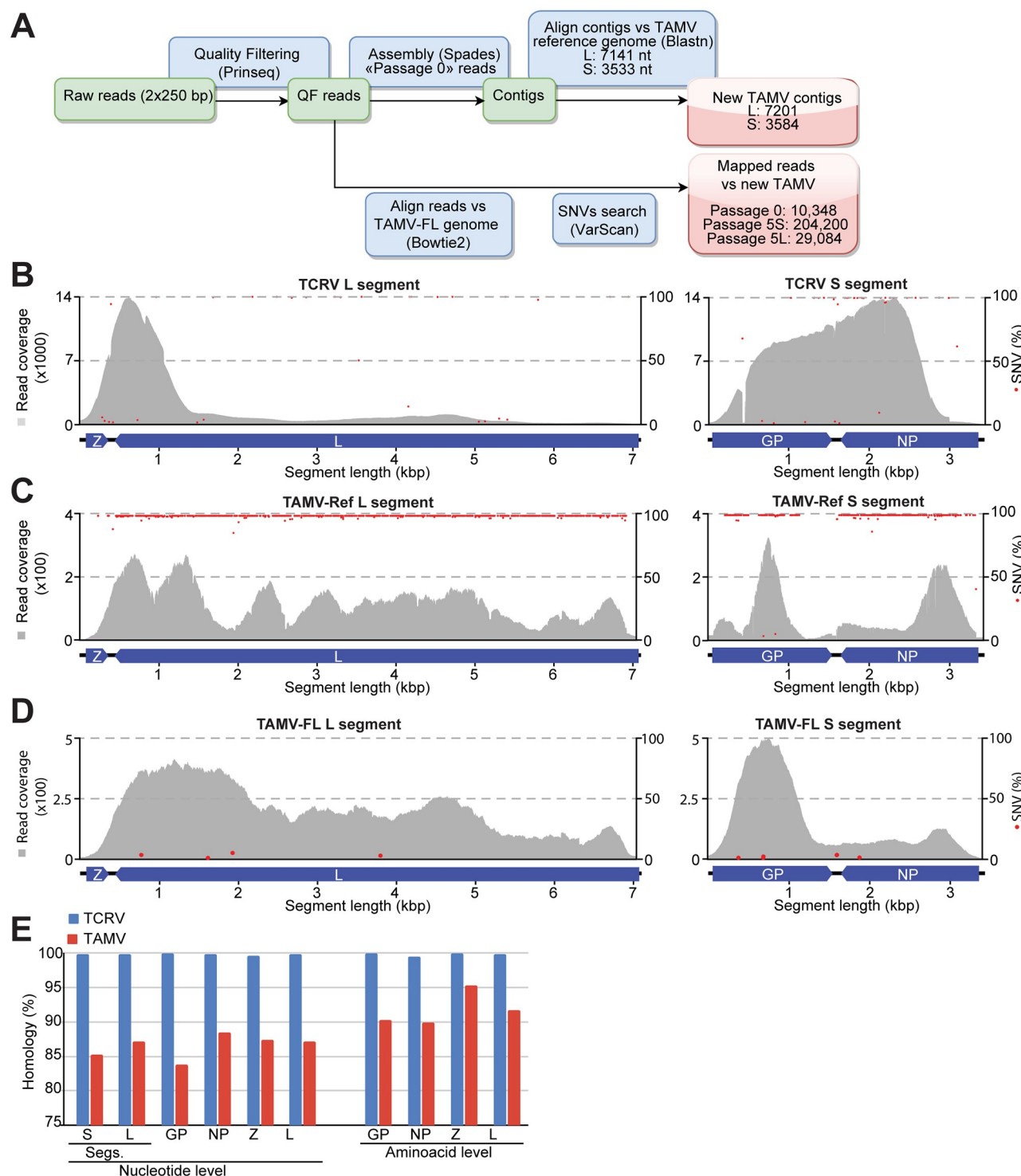

**Fig 1. (A)** Work-flow followed for the NGS analysis. **(B)** TCRV read coverage (Grey area) and relative abundance of SNV (red dots) aligned against TCRV strain 11573 in our tick-derived isolate. **(C)** TAMV read coverage (Grey area) and relative abundance of SNVs (red dots) aligned against TAMV-Ref in tick-derived isolate stock. **(D)** TAMV read coverage (Grey area) and relative abundance of SNVs (red dots) aligned against TAMV-FL in our tick-derived isolate. **(E)** TCRV (red bars) and TAMV-Ref (blue bars) sequence homology at nucleotide and amino acid level in our tick-derived isolate. Alignments were performed with Blastn.

**Table 1. Sequence comparison of TAMV-FL, TAMV-Ref, TAMV- W10777 and AV97140103 strains.** Single Nucleotide Polymorphisms (SNPs) are shown for each TAMV strain. *: Only S segment sequence was available for TAMV AV97140103 strain.

| S Segment | | | | |
|---|---|---|---|---|
| | TAMV W10777 | AV97140103 | TAMV Ref | TAMV FL |
| TAMV-FL (MK500937) | 529 | 517 | 529 | 0 |
| TAMV-Ref (NC_010701.1) [47] | 0 | 640 | 0 | |
| AV97140103 (EU486821.1) * | 640 | 0 | | |
| TAMV-10777 (AF512828.1) [47,49] | 0 | | | |
| L Segment | | | | |
| | TAMV W10777 | TAMV-Ref | TAMV FL | |
| TAMV-FL (MK500936) | 917 | 917 | 0 | |
| TAMV-Ref (NC_010702.1) [47] | 11 | 0 | | |
| TAMV W10777 (EU627614.1) [49] | 0 | | | |

## TAMV-FL utilizes human transferrin receptor-1 more efficiently than TAMV-Ref

The ability of NW mammarenaviruses to use hTfR1 is a critical factor for their potential for zoonotic spillover and ability to cause human disease [21,22,25,26,28,29,50,51]. Although several additional factors are involved during zoonotic events, the importance of hTfR1 in this process prompted us to examine whether TAMV-FL was able to use hTfR1 to infect human cells. Since mammarenavirus cell attachment and entry are both mediated by the viral envelope, we used a bio-contained pseudotype platform to study cell entry mediated by the GP of selected mammarenaviruses. Recombinant vesicular stomatitis virus (rVSV)-derived pseudoviruses (PVs) displaying heterologous viral GPs have become major tools to study cell entry of highly pathogenic emerging viruses [52–59]. We successfully generated PVs decorated with the GPs of JUNV, TAMV-Ref, TAMV-FL, and several other NW mammarenaviruses mentioned in this study. Moreover, we quantified GP incorporation in our PV preparations by calculating the ratio of TAMV GP (HA-tagged) and the VSV matrix protein (VSV-M) by immunoblotting. We found that TAMV-FL PV contained more GP than TAMV-Ref PV (S2A Fig). To further evaluate the impact of the amount of GP incorporated into PV preparations, we produced TAMV-FL PVs using ranged amounts of GP (S2B Fig). We found that, although the amount of GP in PV preparations affects PV titers (S2C Fig), infections performed with inoculums normalized with the detected amount of GP results in comparable PV infectivity (S2D Fig). These results indicate that the amount of GP in our PV does not affect infectivity and further validate the use of VSV-based PVs to compare GP-mediated entry. Furthermore, aiming to perform faithful comparisons between TAMV-Ref and TAMV-FL, minimizing any possible bias of previous virus passage history, for our TAMV-Ref PV production, we *in vitro* synthetized TAMV-Ref GP from publicly available sequence in GenBank NC_010701.1.

We first performed a siRNA knock down of 293T endogenous hTfR1 expression (S3A Fig) and investigated the consequences for GP-mediated cell entry of JUNV, TAMV-Ref and TAMV-FL. Entry of JUNV, which in addition to its rodent host TfR1, *Calomys musculinus*, can utilize hTfR1 as an entry receptor [28], is significantly inhibited by hTfR1 silencing (Fig 2A). Intriguingly, TAMV-FL entry, but not TAMV-Ref, GP-mediated entry is significantly reduced in cells with decreased hTfR1 expression (Fig 2A). To confirm these results, we performed infection assays in airway epithelial A549 cells, which are suitable cellular systems to address possible human infection by inhalation of virus-containing aerosols [16–19], and in hTfR1-partially-depleted cells. To this aim, we used CRISPR/Cas9 to generate A549 cells with

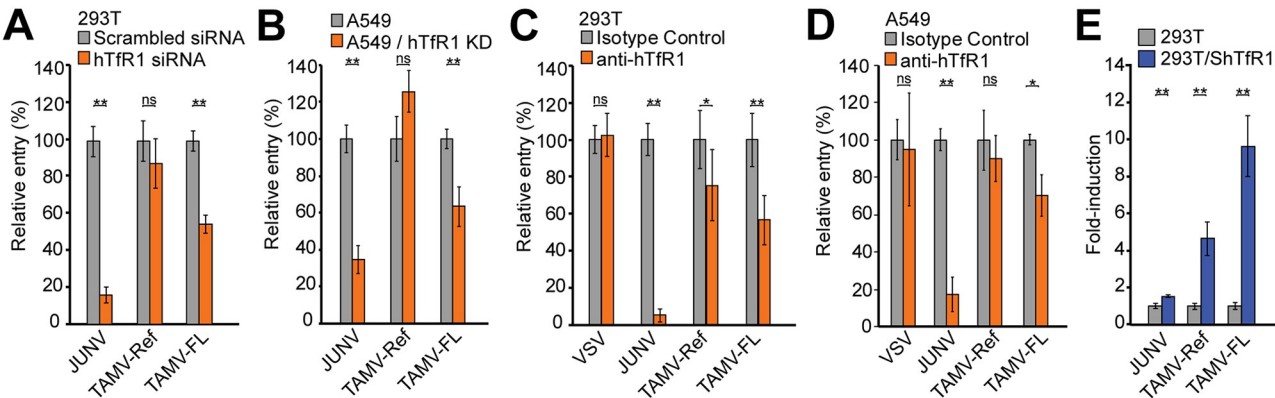

**Fig 2. Transferrin receptor-1 (human and *Sigmodon hispidus* orthologues) dependence of TAMV-FL. (A)** Relative entry of NW arenavirus in siRNA-reduced hTfR1 expression in 293T cells. hTfR1-targeted or non-targeted siRNA siRNAs were transfected to transiently reduce hTfR1 expression or used as reference control of infection (S3A Fig). Error bars represent standard deviations (n = 5). **(B)** Relative entry of NW arenavirus in CRISPR/Cas9-reduced hTfR1 expressing A549 cells. (A549/hTfR1 KD). hTfR1 expression in un-transduced (A549) and A549/hTfR1 KD cells was monitored by FACS (S3B Fig). Error bars represent standard deviations (n = 4). **(C)** Relative infection of NW mammarenavirus in 293T cells in presence of anti-hTfR1 blocking antibody. Reference values were obtained from infections performed in presence of identical concentrations of isotype control. Error bars represent standard deviations (n = 12). **(D)** Relative infection of NW mammarenavirus in A549 cells in presence of anti-hTfR1 blocking antibody. Reference values were obtained from infections performed in presence of identical concentrations of isotype control. Error bars represent standard deviations (n = 3). **(E)** Relative entry of NW mammarenaviruses in *Sigmodon hispidus* (Sh)TfR1 transiently expressing 293T cells. ShTfR1 surface expression was confirmed by FACS analysis at the moment of the infection. (S3E Fig). Error bars represent standard deviations (n = 4). Asterisks in all panels denote statistical significance in ANOVA test. (n.s.: p>0.05; *: p<0.05; **: p<0.01).

a knockdown for hTfR1. Due to observed significant reduction of cell growth rate and survival that could bias our results, we generated A549 cells in which hTfR1 expression was reduced to residual levels (A549/hTfR1 KD) but not completely abrogated, which behaved as parental A549 cells (S3B Fig). Using A549/hTfR1 KD cells, we confirmed that TAMV-FL, compared to TAMV-Ref, is better able to utilize hTfR1 for cell entry (Fig 2B). To validate that the observed hTfR1 dependence was a consequence of specific interactions between the viral GP and hTfR1, we next examined the effect of a monoclonal blocking antibody against hTfR1 on JUNV and TAMV cell entry into 293T and A549 cells, which express high levels of hTfR1 (S3C and S3D Fig). In the presence of the anti-hTfR1 blocking antibody [23,28], the relative entry of JUNV in 293T and A549 cells is inhibited 95% and 82.4%, validating our experimental setup. Confirming previous results, TAMV-FL is inhibited by the anti-hTfR1 antibody more efficiently than TAMV-Ref (Fig 2C and 2D). We next examined the ability of TAMV-Ref and TAMV-FL to use the TfR1 orthologue of the TAMV natural reservoir, *Sigmodon hispidus* (ShTfR1), using 293T cells expressing ShTfR1 (S3E Fig). Whilst JUNV GP-mediated infectivity increase in ShTfR1 expressing cells is negligible, consistent with its inability to use this receptor orthologue [15,21], TAMV-Ref and TAMV-FL entry is increased 4.6- and 9.6-fold, respectively in ShTfR1-transfected 293T cells compared to parental cells (Fig 2E). These data are consistent with the ability of both TAMV-Ref and TAMV-FL to use the TfR1 of the native host species.

## Isolation of replicating TAMV-FL from tick-derived samples

Despite the proven ability of the TAMV-FL GP to support viral entry, it was unclear if the TAMV-FL sequences identified in *A. americanum* ticks represented a replicating virus. A major challenge for the isolation of TAMV-FL was the vast excess of replicating TCRV present in our samples. However, TAMV is capable of exerting an unusually broad non-reciprocal superinfection exclusion of other NW arenaviruses in macaque Vero cells, preventing productive secondary infections of phylogenetically distant species like TCRV [43]. Therefore, we

hypothesized that, if non-reciprocal superinfection exclusion by TAMV is not a cell-type dependent trait, TAMV-FL could outcompete excess of TCRV during serial passages. To this end, we used A549 cells, which are known to be highly permissive for NW mammarenaviruses and have been extensively used in experimental studies with mammarenaviruses [60–63]. While TCRV is known to be sensitive to type I interferon (IFN-I) [63,64], sensitivity of TAMV-FL to IFN-I is unknown. Therefore, we reasoned that an IFN-induced antiviral state could favor TAMV to outcompete TCRV during serial passages. Briefly, A549 monolayers were left untreated or IFN-pretreated and were initially infected with the TCS from the tick-derived sample passaged three times in VeroE6 cells. This inoculum contained predominantly TCRV and 0.58% of TAMV-FL (S1 Table). To allow for multiple rounds of virus infection during each passage, cells were infected at low TCRV multiplicity of infection (MOI; 0.05 PFU/cell) and TCS were collected either after 2 (short passages) or 5 days (long passages). Infectious TCRV titers from collected TCS were determined by immunofocus assay (IFA) using monoclonal antibody MA03, which recognizes TCRV NP but not TAMV NP [65]. Due to lack of highly specific TAMV antibodies, it was impossible to monitor TAMV infectious viral titers along the passages. During serial passaging, we observe a progressive drop in infectious TCRV titers, resulting in undetectable levels between passages 2 and 6 (Short passages format) or 2 and 4 (Long passages format), corresponding to a total of 96–288 hpi and to 240–480 hpi, respectively (Fig 3A). To examine if the loss of infectivity of TCRV in TCS correlates with the presence of TAMV-FL, we infected fresh VeroE6 cells with undiluted TCS from passage 6 (IFN-pretreated cells, short format passage) and 48 hpi, fixed the cells and examined them by IFA. In order to distinguish between TCRV and TAMV infected cells, we used the aforementioned MA03 antibody and the broadly cross-reactive IC06 antibody, which recognizes both TCRV and TAMV NP [65]. Consistent with undetectable TCRV titers in TCS, cells infected with TCS of passage 6 show no detectable staining for TCRV NP above background (Fig 3B). In contrast, probing with the cross-reactive IC06 antibody reveals a cytosolic staining pattern characteristic of mammarenavirus NP expression, demonstrating the presence of a replicating NW arenavirus other than TCRV (Fig 3B). To verify the genetic identity of the selected virus, we examined TCS from passage 5 in IFN-pretreated cells (short and long passage format; TAMV-FLp5s and TAMV-FLp5l respectively) by NGS as depicted in Fig 1A. The obtained sequence data confirms the extinction of TCRV and selection of TAMV-FL that represents 100% of all arenavirus reads in both samples, TAMV-FLp5s and TAMV-FLp5l (Fig 3C), demonstrating the presence of replicating TAMV-FL in our tick-derived samples. The NGS analysis allowed us to assemble the entire TAMV-FL genome in both samples. Consistent with mammarenaviruses quasispecies dynamics [14,30,31], the alignment of TAMV-FLp5s and TAMV-FLp5l (Fig 3D) against TAMV-FL genome reveals 43 single nucleotide variants (SNVs) in both segments of TAMV-FL genome (Table 2).

## Acquisition of non-synonymous mutations upon short-term TAMV-FL passage

At the amino acid level, the 43 SNVs found in TAMV-FL5s and TAMV-FLp5l upon serial passage result in 18 amino acid substitutions (Table 2).

In TAMV-FLp5s quasispecies, the frequency of the GP1 variants N151K and D156N increase from 0.11% and 0% to 18.75% and 20.03%, respectively ($p < 10^{-12}$ in both cases), with respect to TAMV-FL. In TAMV-FLp5l, the substitution N151K enriches to 32.39% ($p < 10^{-12}$), while D156N only accumulates to 0.33% ($p > 10^{-6}$) (Table 2). While 569 and 623 reads, out of a total of 2997, harbor T540A (N151K) and G553A (D156N) substitutions, respectively, only 2 reads contain both mutations ($p < 0.01$, $\chi^2$ value = $6.3 \times 10^{-12}$), suggestive of a strong

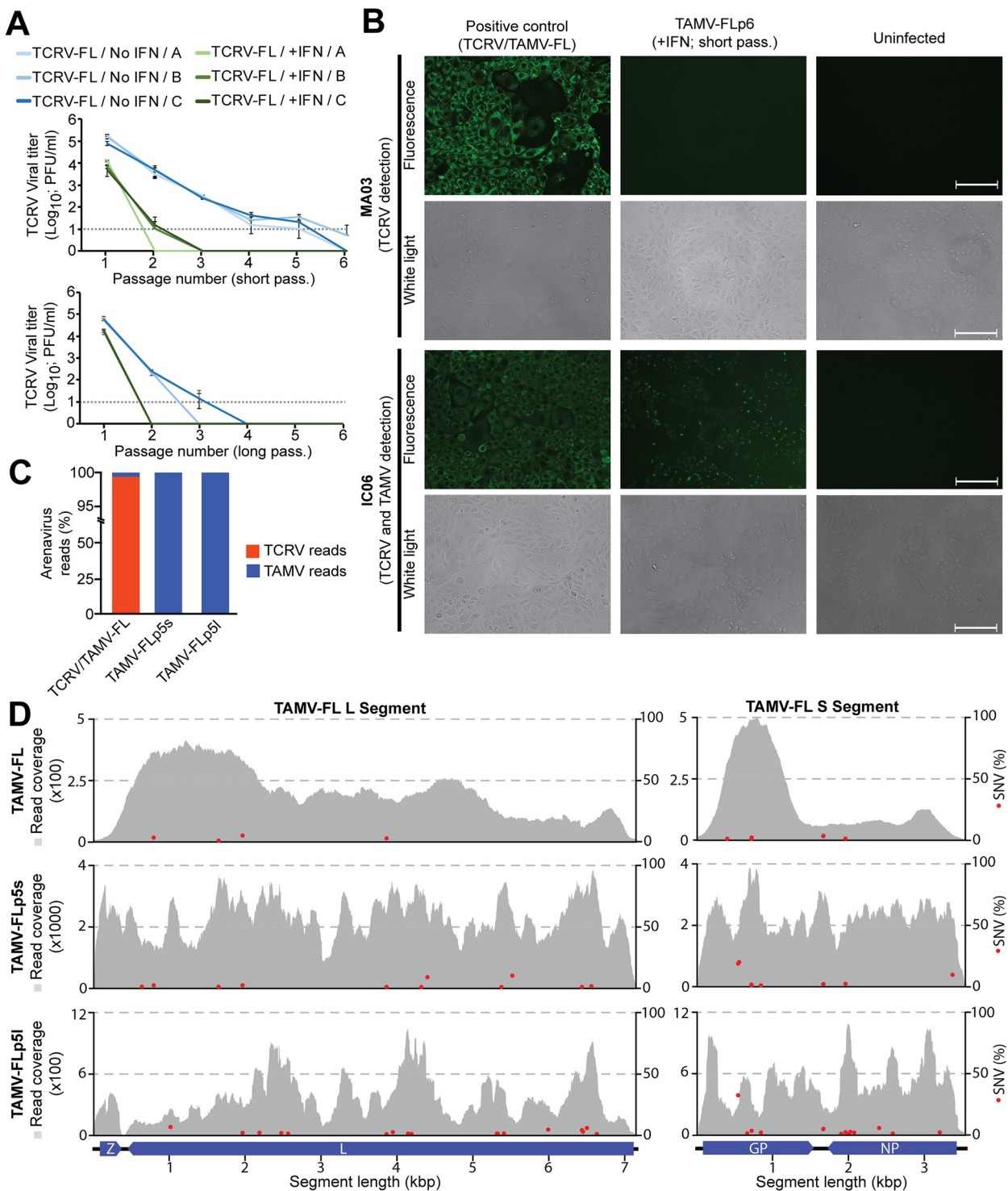

**Fig 3. TAMV isolation from tick-derived sample. (A)** TCRV viral titers along short (2 days per passage) and long (5 days per passage) passages in A549 cells pretreated or not with 100 IU/ml of IFN. Grey dotted line represent the detection limit of the experiment. Error bars represent standard deviations (n = 4). Biological replicas (A, B and C) are shown independently. **(B)** Identification of TAMV by IFA in VeroE6 infected cells (48h after infection) with TCS from passage 6 (short format) in presence of IFN (replica A). MA06 antibody recognizes TCRV NP and IC06 antibody recognizes TCRV and TAMV NPs. Scale bars represent 125 μm. **(C)** Relative abundance of TCRV and TAMV-FL reads obtained by NGS analysis of tick-derived original sample (passage 0, TCRV/TAMV-FL) and of TCS from passage 5 in presence of IFN of short (TAMV-FLp5s) or long (TAMV-FLp5l) passages. **(D)** TAMV-FL, TAMV-FLp5s and TAMV-FLp5l read coverage (Grey area) and relative abundance of SNV (red dots) of aligned against TAMV-FL sequence.

**Table 2. Quasispecies composition of TAMV-FLp0, TAMV-FLp5FLp5s and TAMV-FLp5FLp5l.** α indicates SNV identifier, consisting of the viral gene, the observed substitution, and position of the substitution. β indicates relative abundance for each particular SNV. δ indicates the amino acid change resulting from the SNV (empty fields denote synonymous or insertion/deletion mutations). * stands for $p<10^{-6}$ and ** for $p<10^{-12}$. SNVs GP540A and GP553A are highlighted in bold.

| Identifier[α] | Relative Frequency[β] | | | Coverage | | | Amino acid Change[δ] |
|---|---|---|---|---|---|---|---|
| | TAMV-FL | TAMV-FLp5s | TAMV-FLp5l | TAMV-FL | TAMV-FLp5s | TAMV-FLp5l | |
| L / T634C | 0% | 1.34% | 0% | 296 | 1339 | 115 | |
| L / T790C | 3.58% | 2.47% (*) | 0% | 363 | 1335 | 110 | |
| L / T1012C | 0% | 0.15% | 6.83% | 360 | 2724 | 249 | I2034M |
| L / A1645G | 1.17% | 1.32% (*) | 0.78% | 341 | 3328 | 386 | |
| L / C1961+A | 5.25% | 2.52% (**) | 2.06% | 324 | 2900 | 291 | |
| L / G2184T | 0% | 0.38% | 2.17% | 233 | 2861 | 600 | L1644I |
| L / G2471A | 0% | 0% | 1.99% | 219 | 2504 | 653 | T1548M |
| L / C2561T | 0.5% | 0.12% | 1.49% | 199 | 2437 | 737 | S1518N |
| L / T3858-A | 3.01% | 1.17% (*) | 1.02% | 166 | 2398 | 295 | |
| L / G3941A | 0% | 0.06% | 2.64% | 170 | 3212 | 416 | T1048I |
| L / A4141G | 0% | 0% | 1.48% | 181 | 2758 | 1013 | |
| L / A4186G | 0% | 0% | 1.17% | 184 | 2668 | 768 | |
| L / A4315G | 0% | 1.04% (*) | 0% | 194 | 2891 | 752 | |
| L / C4396T | 0.95% | 9.11% (**) | 0% | 211 | 2305 | 539 | |
| L / A5311G | 0% | 0.2% | 1.56% | 118 | 2514 | 513 | |
| L / G5325T | 0% | 0.17% | 1.33% | 123 | 2933 | 526 | L597M |
| L / T5371+A | 0.9% | 1.05% (*) | 0% | 111 | 2946 | 446 | |
| L / C5404T | 0% | 0.06% | 1.6% | 95 | 3286 | 438 | |
| L / C5515T | 0% | 10.35% (**) | 0% | 101 | 2444 | 167 | |
| L / T5987C | 0% | 0.06% | 4.76% | 91 | 1644 | 273 | Y376C |
| L / T6431C | 0% | 1.09% (*) | 4.35% (*) | 60 | 3033 | 506 | H228P |
| L / T6445C | 0% | 0.07% | 3.09% | 59 | 2872 | 517 | |
| L / G6497A | 0% | 0.03% | 5.97% (**) | 65 | 3274 | 737 | A206V |
| L / T6556C | 0% | 1.77% (**) | 0.12% | 76 | 3382 | 848 | |
| L / C6628T | 0% | 0.03% | 1.15% | 94 | 3208 | 609 | M162I |
| GP / A397G | 1.27% | 0.03% | 0.4% | 628 | 3339 | 248 | I104V |
| **GP / T540A** | **0.11%** | **18.75% (**)** | **32.39% (**)** | **874** | **3094** | **284** | **N151K** |
| **GP / G553A** | **0%** | **20.03% (**)** | **0.31%** | **903** | **3265** | **319** | **D156N** |
| GP / G659A | 0.75% | 0.33% | 1.42% | 937 | 5099 | 564 | R191K |
| GP / T714+A | 1.21% | 1.7% (**) | 0.33% | 991 | 5530 | 306 | |
| GP / A717G | 2.34% (*) | 1.09% (**) | 3.5% | 983 | 5580 | 314 | |
| GP / G839+T | 0.52% | 1.1% (*) | 2.07% | 954 | 3983 | 193 | |
| IGR (S) / T1658A | 3.45% | 2.05% (**) | 4.78% | 116 | 4780 | 251 | |
| IGR (S) / T1661A | 3.67% | 2.19% (**) | 5.11% | 109 | 4620 | 235 | |
| NP / C1893T | 0% | 0.1% | 1.08% | 140 | 3024 | 647 | |
| NP / T1950C | 1.48% | 2.35% (**) | 2.34% | 135 | 2899 | 768 | |
| NP / T1997C | 0% | 0.46% | 1.05% | 128 | 3926 | 1047 | I473V |
| NP / A2013G | 0% | 0.03% | 2.64% (*) | 130 | 3893 | 983 | |
| NP / A2065G | 0% | 0.52% | 2.05% | 127 | 3645 | 684 | V450A |
| NP / T2393C | 0% | 0% | 5.74% (*) | 168 | 4378 | 470 | K341E |
| NP / T2572C | 0.76% | 0.4% | 1.21% | 132 | 4539 | 829 | E281G |
| NP / A3189G | 0% | 0.37% | 2.22% | 146 | 3784 | 496 | |
| NP / T3357C | 0% | 9.73% (**) | 0% | 43 | 3742 | 460 | |

evolutionary disadvantage against the TAMV-FL double mutant T540A/G553A (N151K/D156N). The remaining non-synonymous SNVs show only limited changes in relative abundance during our serial passage experiment and were not further analyzed.

## Mutations acquired in TAMV-FLp5s/l GP lie outside the predicted transferrin receptor 1 -binding interface

Recent structures of the attachment-mediating GP1 glycoprotein from the clade D WWAV have revealed structural diversity within the GP1 of TfR1-tropic viruses [25,66], principally within GP1 loop regions that are likely involved in TfR1 recognition [26]. Such structural variation across NW mammarenaviruses with shared receptor-tropism may indicate diverse modes of TfR1 recognition [67]. WWAV-AV96, which was associated with fatal human illness [20], utilizes hTfR1 for viral entry [25,51]. Interestingly, sequence alignment of WWAV and TAMV strains reveals common D to N substitutions at positions 156 and 154 of the GPs in TAMV-FL and WWAV-AV96, respectively (Fig 4A). We therefore thought to assess the potential functional effects of the TAMV-FL GP1 N151K and D156N substitutions, enriched during serial passaging of TAMV-FL in human cells. To that end, we performed structure-based mapping, utilizing the crystal structure of the closely related WWAV GP1 (Protein Data Bank [PDB] code 6HJ5) as a proxy. Our model localizes the N151K and D156N substitutions to the helical face of the molecule, distal to the putative TfR1-binding interface [26] (Fig 4B). To further assess the spatial relationships of these substitutions in the context of mature GP, a model of the trimeric clade D NW mammarenaviral GP was constructed based upon superposition of WWAV GP1 onto the structure of the LASV GP ectodomain (PDB: 5VK2), as has been proposed previously for MACV GP [7]. Our model places both amino acid positions, N151K and D156N, near the trimeric axis of the spike, in a sterically constrained environment that would likely be inaccessible to large protein ligands, such as TfR1 (Fig 4C and 4D). Furthermore, superposition of the C-terminal region of LASV GP1 (E228–S255$^{LASV}$), which is truncated in all available isolated NW arenavirus GP1 structures, suggests that both N151K and D156N substitutions may be wholly or partially buried in the context of full-length GP1 within mature GP. Given the spatial distinction from the putative TfR1-binding site and their likely position near the sterically constrained trimeric axis of GP, the N151K and D156N substitutions are unlikely to modulate TfR1 utilization by directly affecting the GP1-TfR1 interface. We thus decided to conduct functional analyses in order to clarify the functional effects of these mutations.

## Single N151K and D156N substitutions increase hTfR1 entry dependence of TAMV-FL

To assess the possible impact of N151K and D156N changes on hTfR1 usage, we introduced the aforementioned substitutions into TAMV-FL GP1 and produced the corresponding PVs (TAMV-FL N151K, TAMV-FL D156N and TAMV-FL DM, respectively) as well as WWAV-Ref and WWAV-AV96 PVs. Quantification of GP incorporated into the PVs indicates that TAMV-FL N151K, TAMV-FL D156N and TAMV-FL DM PVs contained less GP than the TAMV-FL (S4 Fig). Next, we infected A549 and A549/hTfR1 KD cells and found that the introduction of either substitutions, N151K or D156N, increases hTfR1 usage compared to the parental TAMV-FL, whilst the double mutant exhibits decreased hTfR1 usage, comparable to TAMV-FL. This suggests that these mutations cancel out their individual functional advantage when combined (Fig 5A). To confirm these results, we infected 293T and A549 cells in presence of an anti-hTfR1 blocking antibody (Fig 5B and 5C, respectively), and observed that either mutation, N151K or D156N, increases hTfR1 usage of TAMV-FL GP, whereas this effect

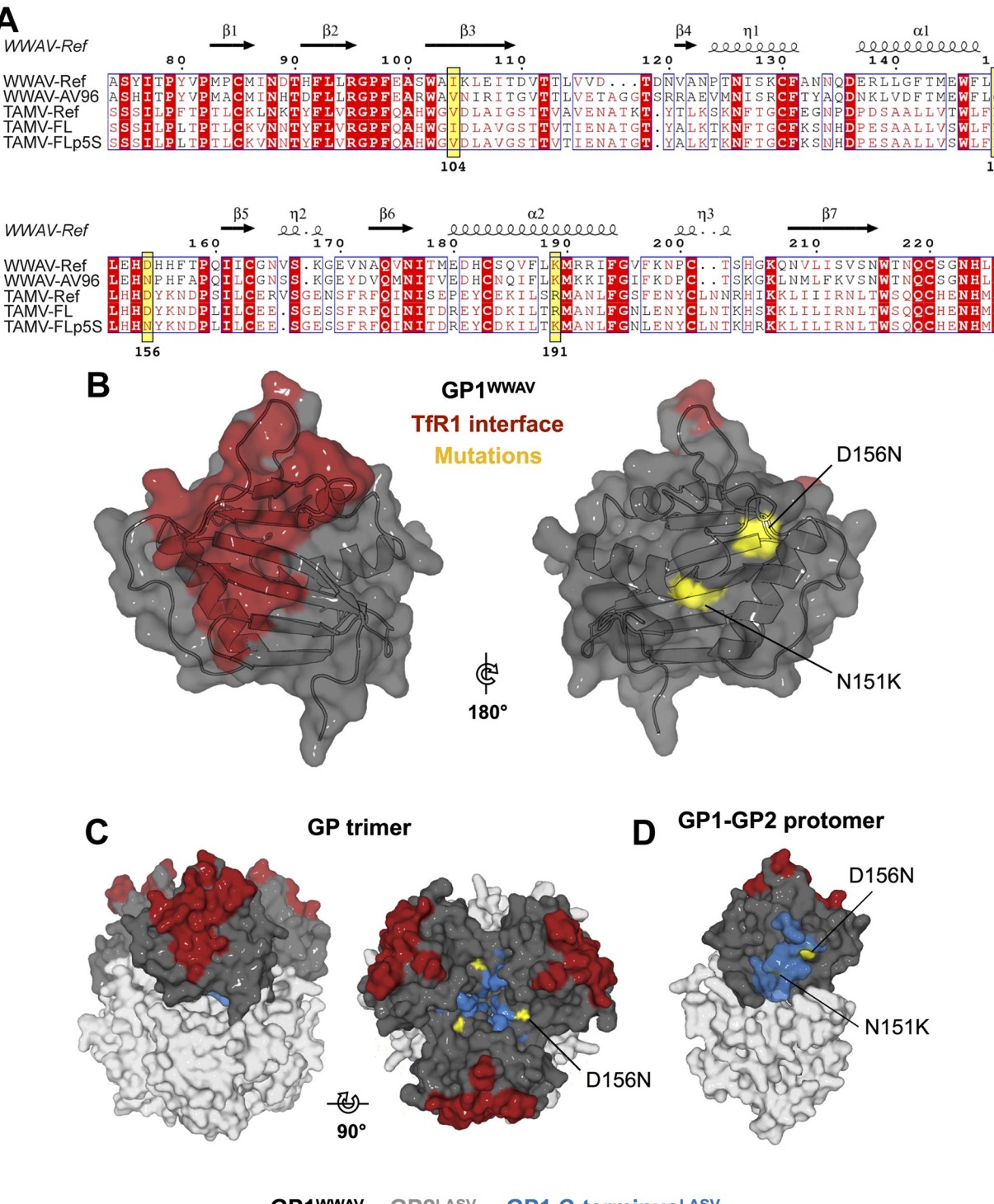

**Fig 4. Structure-based mapping of acquired TAMV mutations. (A)** Amino acid sequence alignment of NW clade D GP1s, numbered according to WWAV-Ref. Polymorphisms common to TAMV and WWAV are highlighted in yellow and numbered in accordance with TAMV-FL numbering. Secondary structural elements of WWAV GP1 are illustrated above the sequences. Alpha (α) and 3₁₀ (η) helices are shown as coils, and β-sheets (β) as arrows. Conserved residues are shown in red boxes and semi-conserved residues are shown in red font within a blue box. Alignments were displayed using ESpript (http://espript.ibcp.frhttp://espript.ibcp.fr) [109]. **(B)** Amino acid substitutions (yellow) acquired by TAMV_FL map to the putatively trimeric axis-facing helical region of GP1, distal to the expected TfR1 receptor binding site (red). Features are mapped onto the structure of WWAV GP1. **(C)** Localization of TAMV mutants in the context of a composite model of trimeric GP, excluding SSP and transmembrane domains. The GP2 subunit from the trimeric LASV GP structure is shown in light grey, with GP1 from WWAV (colored as in panel B) in place of LASV GP1. The C-terminal region of LASV GP1, which is not present in the structure of WWAV GP1, is shown overlaid in blue. **(D)** View of the trimeric axis-facing side of a composite GP1-GP2 protomer, colored as in panel B.

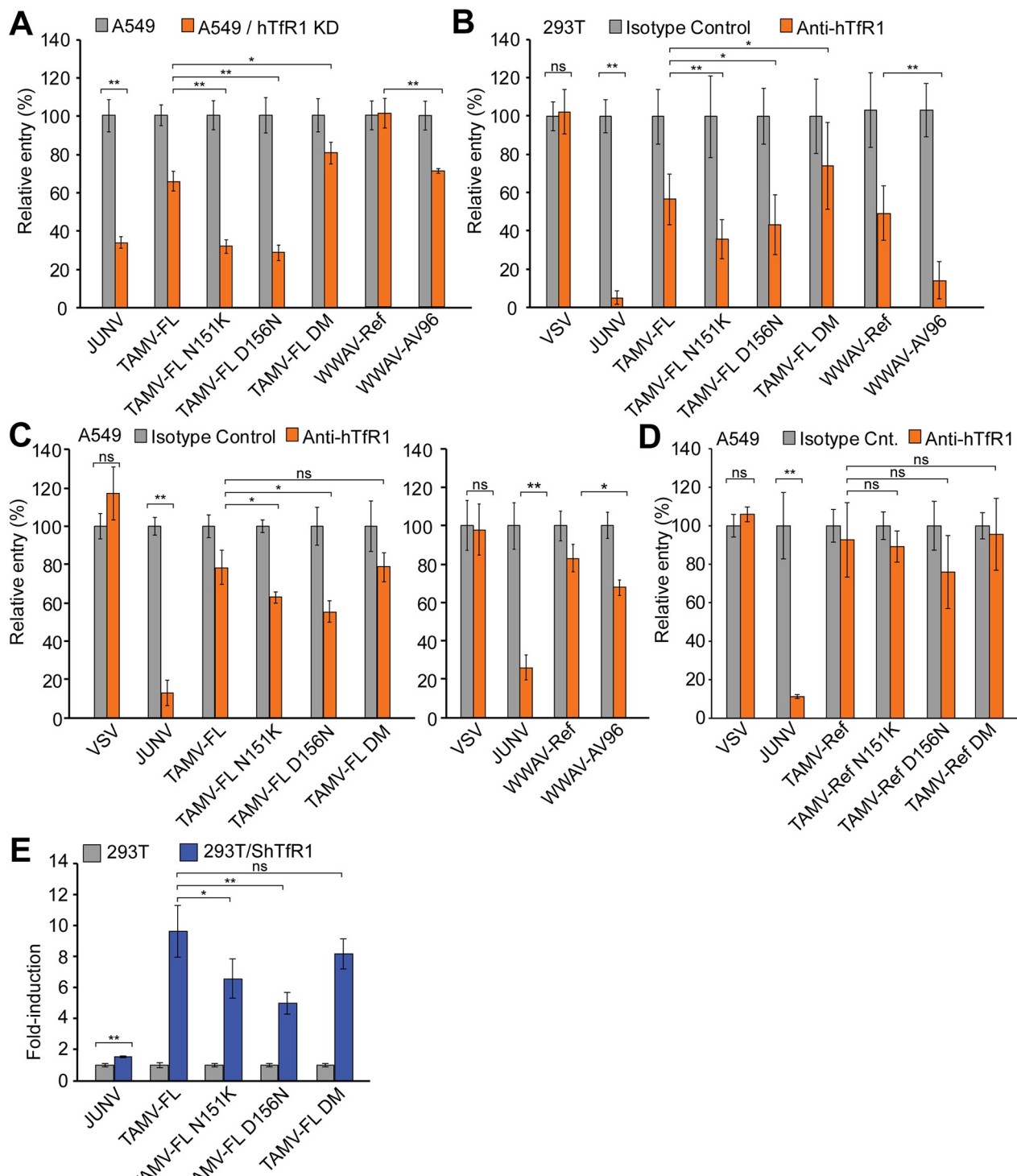

**Fig 5. Transferrin receptor-1 (human and *Sigmondon hispidus* orthologues) use of TAMV-FL mutants. (A)** Relative entry of NW mammarenavirus in CRISPR/Cas9-reduced hTfR1 expressing A549 cells (A549/hTfR1 KD). hTfR1 expression in non-transduced (A549) and hTfR1hA549/TfR1 KD cells was monitored by FACS (S3B Fig). Error bars represent standard deviations (n = 4). Antibody perturbation against endogenous hTfR1 with specific blocking antibody in **(B)** 293T cells and **(C)** A549 in infections of TAMV-FL and TAMV-FL mutants. Reference values were obtained from infections performed in presence of identical concentrations of isotype control. Error bars represent standard deviations (panel B, n = 12; panel C, n = 3) **(D)** Antibody perturbation experiments, with anti-hTfR1 specific antibody or isotype control, in infections with TAMV-Ref and TAMV-Ref mutants in A549 cells. Error bars represent standard deviations (n = 3). **(E)** Relative entry of NW mammarenavirus on *Sigmodon hispidus* (Sh)TfR1 transiently expressing 293T cells. ShTfR1 surface expression was monitored by FACS analysis at the moment of the infection (S3E Fig). Error bars represent standard deviations (n = 4). Asterisks in all panels denote statistical significance in ANOVA test (ns: p>0.05; *: p<0.05; **: p<0.01).

was abrogated if both mutations are simultaneously present in the same genome. These results also confirmed that WWAV-AV96 utilizes hTfR1 more efficiently than WWAV-Ref (Fig 5B and 5C) [25,51].

We then investigated whether the GPC genetic backbone is important in determining the effect of the N151K and D156N substitutions upon hTfR1 binding. To this end, we introduced either N151K, D156N, or both mutations, in TAMV-Ref GPC backbone, produced the corresponding PVs (TAMV-Ref N151K, TAMV-Ref D156N and TAMV-Ref DM, respectively) and infected A549 cells in presence of the anti-hTfR1 blocking antibody. We found that the mutations, either alone or in combination, had a negligible impact on hTfR1 dependence in the TAMV-Ref genetic backbone (Fig 5D).

Furthermore, we investigated the ability of TAMV-FL variants to bind ShTfR1. We infected 293T cells transiently expressing ShTfR1 (S3E Fig) with parental TAMV-FL, and TAMV-FL mutant PVs. Either individual substitutions, N151K or D156N, reduces the ability of GP1 to bind ShTfR1, but the changes antagonize each other when present together (TAMV-FL DM), leading to a ShTfR1 usage comparable to that of the parental TAMV-FL (Fig 5E).

## Substitutions N151K and D156N increase affinity for heparan sulfate proteoglycans

Efficient heparan sulfate binding has been reported in several viruses [35,36,40,68,69]. Amino acid changes N151K and D156N lead to charge alterations in the viral GP, and therefore may modulate the HSPG binding and promote viral entry. Therefore, we examined the infectivity of TAMV-Ref, TAMV-FL, TAMV-FL N151K, TAMV-FL D156N and TAMV-FL DM PVs, along with WWAV-Ref and WWAV-AV96 PVs, in presence of increasing concentrations of competing heparin (Fig 6A and S5A Fig). As reference, we used Ebola virus (EBOV) PV, which is known to bind HSPG with high affinity [40]. Entry of both TAMV-FL mutant PVs, N151K and D156N, is more strongly inhibited by heparin than entry of the parental TAMV-FL, indicating increased ability to bind HSPG. Although virtually absent in any natural TAMV-FLp5 population analyzed, the double mutant N151K/D156N TAMV-FL PV shows the highest binding capacity to HSPG among the viruses tested. Interestingly, WWAV-AV96 is also more inhibited in the presence of heparin than WWAV-Ref (Fig 6A). To validate these results, we also performed PV infections in A549 cells treated with heparinase III, which cleaves and removes the polysaccharide chains from the HSPG protein component (Fig 6B and S5B Fig). The results confirm an increased HSPG binding capacity of TAMV-FL N151K, D156N and DM variants as well as WWAV-AV96 compared to TAMV-FL and WWAV-Ref, respectively.

In order to evaluate the effect of the genetic backbone on the HSPG binding, we used TAMV-Ref and TAMV-Ref mutants to infect A549 cells in the presence of different concentrations of heparin, and monitored relative entry (Fig 6C). The results show that TAMV-Ref N151K, TAMV-Ref D156N and TAMV-Ref DM exhibit increased HSPG dependence, comparable to the analogous TAMV-FL variants.

## N151K and D156N substitutions delay GP-mediated endosomal escape

After receptor-mediated endocytosis, NW mammarenaviruses are delivered to late endosomes [70]. The transition from early to late endosomal compartments is accompanied by progressive acidification of the endosomal lumen, resulting in a pH gradient that serves as guidance cue for membrane fusion triggering (Fig 7A). The transmembrane GP2 undergoes low pH-induced conformational changes that include transition from a metastable pre-fusion state to an energetically favorable post-fusion state with a six-helix bundle architecture reminiscent to

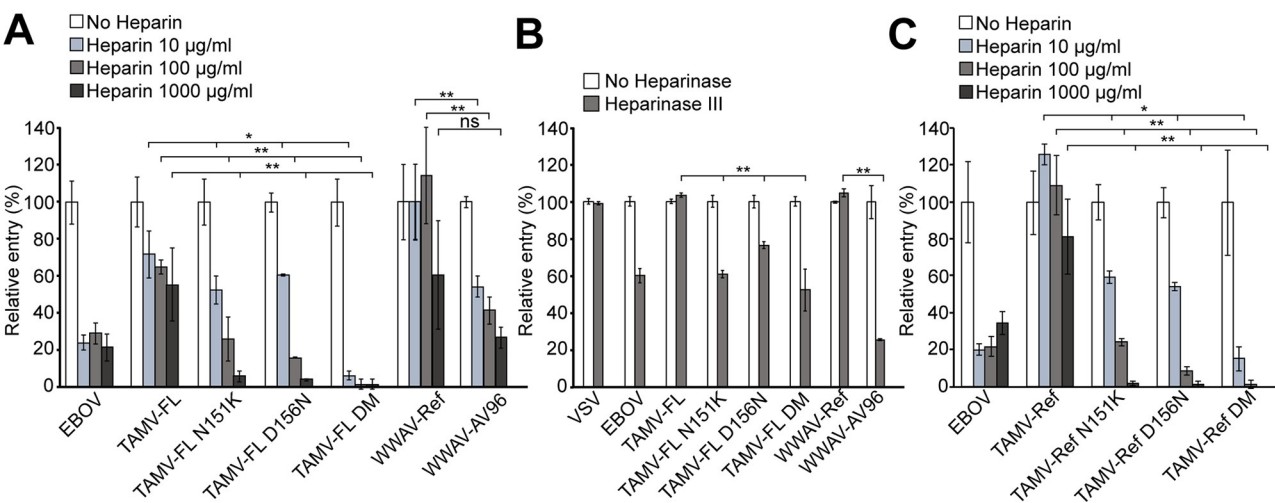

**Fig 6. HSPG binding modulation by N151K and D156N substitutions. (A)** Relative entry of TAMV-FL mutants in infections performed in presence of increasing concentrations of heparin. Error bars represent standard deviations (n = 4). **(B)** Relative entry of PV infections performed in heparinase III-A549-treated cells. Error bars represent standard deviations (n = 4). **(C)** Relative entry of TAMV-Ref mutants in infections performed in presence of increasing concentrations of heparin. Error bars represent standard deviations (n = 4). Asterisks in all panels denote statistical significance in ANOVA test (ns: p>0.05; *: p<0.05; **: p<0.01).

other class I viral fusion protein [71,72]. Given that N151K and D156N map to the sterically constrained axis of GP (Fig 4C), we hypothesized that the substitutions may affect the overall conformational stability of GP and hence modulate pH-dependent GP2-mediated fusion (Fig 4C and 4D). To interrogate this hypothesis, we investigated the effects of N151K and D156N on the pH-triggered GP-mediated fusion activity of TAMV-FL in the context of productive PV entry into A549 cells. Ammonium chloride rapidly depletes the endosomal pH gradient in a concentration-dependent manner, without causing overall cytotoxicity and preventing pH-mediated viral exit from endosomal compartments [73,74]. Thus, sensitivity to ammonium chloride is inversely proportional to the viral fusion pH, with late fusing viruses showing higher sensitivity. To evaluate pH-dependent fusogenic activity of TAMV-FL mutants, we raised the endosomal pH by adding increasing concentrations of ammonium chloride to cells and monitored productive PV entry. To validate our system, we established dose-response curves of ammonium chloride for VSV, which is known to escape from early endosomes at pH >6 [75] and JUNV, which escapes from late endosomes at pH <5.5 [76]. As expected, based on the lower fusion pH, JUNV PV shows higher sensitivity to ammonium chloride than VSV PV ($IC_{50}$ values 1.07 mM and 2.4 mM, respectively) (Fig 7B and Table 3).

Ammonium chloride dose-response curves for TAMV-Ref and TAMV-FL PV reveal similar $IC_{50}$ values of 1.34 mM and 1.54 mM, respectively (Fig 7B and Table 3). Interestingly, introduction of either the substitution N151K or D156N into TAMV-FL GP1 significantly increases ammonium chloride sensitivity, resulting in $IC_{50}$ values of 1.14 mM and 0.99 mM, respectively (Fig 7B and Table 3). The presence of both changes (TAMV-FL DM) yields the lowest $IC_{50}$ (0.62 mM) of all tested PVs. WWAV-AV96 PV also shows reduced ammonium chloride sensitivity and lower $IC_{50}$ than WWAV-Ref PV (1.04 mM and 1.65 mM, respectively). Although our ammonium chloride titration assay is semi-quantitative in nature, it nonetheless evidences the ability of the N151K and D156N substitutions to modulate fusion pH *in vitro*.

The timed gradual decrease in pH delimits the endosomal conditions available for viral escape and release into the cytoplasm (Fig 7A). Hence, under the assumption of identical participating host factors, the timing of sensitivity to ammonium chloride is a proxy for the timing

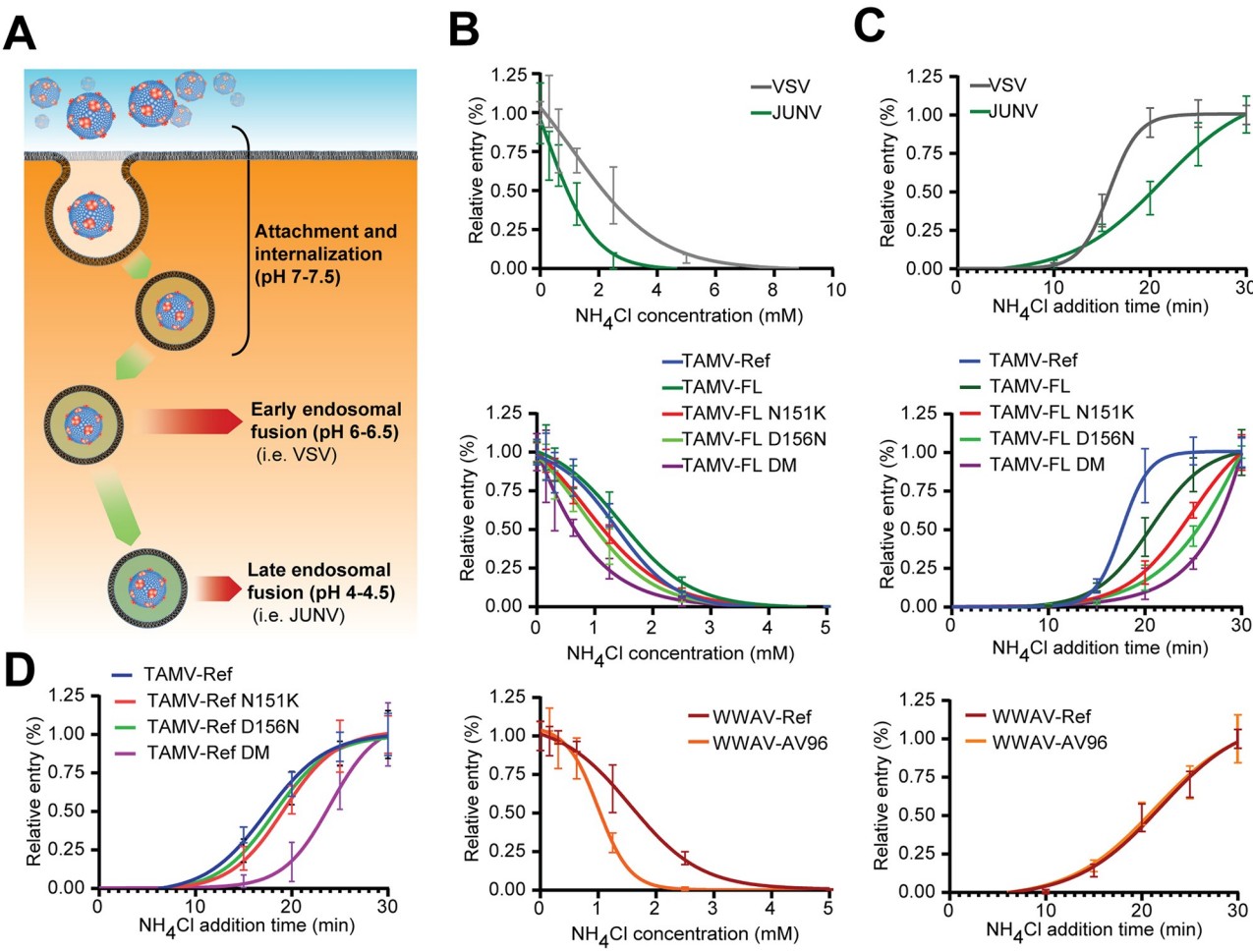

**Fig 7. Endosomal escape of clade D NW mammarenavirus. (A)** Schematic virus internalization, compartment acidification, and pH-dependent endosomal escape. **(B)** Ammonium chloride sensitivity of TAMV-FL mutants and WWAV strains. PV infections were performed in presence titrated ammonium chloride concentrations (0–5 mM). Error bars represent standard deviations (n = 6). **(C)** Endosomal escape of TAMV-FL mutants and WWAV strains. At indicated time points, endosomal exit was prevented with 20 mM of ammonium chloride. Error bars represent standard deviations (n = 3). **(D)** Endosomal escape of TAMV-Ref mutants. Error bars represent standard deviations (n = 7).

**Table 3. Ammonium chloride sensitivity ($IC_{50}$) of PV of VSV, JUNV and clade D NW arenaviruses.** Standard deviations are indicated between parentheses (n = 6). Asterisks (**) indicate statistical significance (p<0.01) in ANOVA test when compared to TAMV-Ref (For TAMV-FL), TAMV-FL (For TAMV-FL mutants) or WWAV-Ref (For WWAV-AV96).

| Virus | $IC_{50}$ (±S.D.) (mM) |
|---|---|
| **VSV** | **2.4** (±0.69) |
| **JUNV** | **1.07** (±0.26) |
| **TAMV-Ref** | **1.34** (±0.09) |
| **TAMV-FL** | **1.54** (±0.09)** |
| **TAMV-FL N151K** | **1.14** (±0.07)** |
| **TAMV-FL D156N** | **0.99** (±0.17)** |
| **TAMV-FL DM** | **0.62** (±0.16)** |
| **WWAV-Ref** | **1.65** (±0.28) |
| **WWAV-AV96** | **1.04** (±0.09)** |

**Table 4. Half-escape time of PV of VSV, JUNV and clade D NW arenaviruses.** Standard deviations are indicated between parentheses (n = 3). Asterisks (**) indicate statistical significance (p<0.01) in ANOVA test when compared to TAMV-FL.

| Virus | Half-escape time (±S.D.) (min) |
|---|---|
| VSV | 15.55 (±0.21) |
| JUNV | 20.52 (±1.46) |
| TAMV-Ref | 17.57 (±0.8) |
| TAMV-FL | 20.28 (±0.6) |
| TAMV-FL N151K | 23.61 (±0.32)** |
| TAMV-FL D156N | 25.4 (±0.89)** |
| TAMV-FL DM | 26.95 (±0.38)** |
| WWAV-Ref | 21.09 (±1.08) |
| WWAV-AV96 | 20.91 (±0.56) |

of endosomal escape. In order to determine the half-escape time of selected clade D NW mammarenaviruses, we added ammonium chloride to TCS at different time points after PV infection, preventing any pH-dependent exit from endosomal compartments. VSV and JUNV PV exit from endosomal compartments approximately 15.5 and 20.5 minutes after infection, respectively (Fig 7C and Table 4). TAMV-FL N151K and TAMV-FL D156N exhibit delayed endosomal escape kinetics and higher half-escape times (23.6 and 25.4 minutes) compared to the parental TAMV-FL, which has a half-escape time of 20.3 minutes. (Fig 7C and Table 4) Endosomal escape of TAMV-FL DM is delayed to a half-escape time of approximately 27 minutes after infection. Intriguingly, despite the absence of significant differences in ammonium chloride sensitivity, TAMV-Ref escapes earlier than TAMV-FL (Fig 7B and 7C, and Tables 3 and 4). WWAV-AV96 shows no delay in half-escape time when compared to WWAV-Ref, whereas the differences in ammonium chloride sensitivity are substantial (Fig 7B and 7C, and Tables 3 and 4). These apparent discrepancies suggest the possible participation of additional host factors during viral entry such as intracellular receptors that have been described for other emerging viruses, including the OW mammarenaviruses LASV and Lujo virus [52,53,55,77]. In summary, these results indicate that N151K and D156N substitutions delay viral release, lead to release from later, and more acidified endosomal compartments.

Next, we examined the impact of N151K and D156N on endosomal escape in the TAMV-Ref GPC genetic backbone. When introduced in TAMV-Ref GPC, N151K and D156N substitutions delay the endosomal escape by 1.7 and 0.8 minutes, respectively, while when introduced in TAMV-FL GP1, N151K and D156N substitutions delay the endosomal escape by 3.3 and 5.1 minutes, respectively (Fig 7C and 7D, and Tables 4 and 5). Incorporation of both mutations results in comparable endosomal escape delay in both TAMV-FL and TAMV-Ref

**Table 5. Half-escape time of PV of TAMV-Ref and TAMV-Ref mutants.** Standard deviations are indicated between parentheses (n = 7). Asterisks (**) indicate statistical significance (p<0.01) in ANOVA test when compared to TAMV-Ref (**: p<0.01).

| Virus | Half-escape time (±S.D.) (min) |
|---|---|
| VSV | 15.03 (±0.79) |
| JUNV | 19.47 (±2.17) |
| TAMV-Ref | 17.55 (±0.51) |
| TAMV-Ref N151K | 19.28 (±1.11)** |
| TAMV-Ref D156N | 18.36 (±0.87) |
| TAMV-Ref DM | 23.85 (±1.12)** |

backbones (6.6 and 6.3 minutes, respectively). Taken together, the TAMV-FL GP1 substitutions, N151K and D156N, not only impact attachment factor usage, but also fusion kinetics and subcellular fusion compartments of the virus, suggesting that these mutations may be adaptive.

## Discussion

The known geographic distribution of mammarenaviruses is limited to that of their rodent reservoir species [12]. The identification of TCRV in host-seeking *Amblyomma americanum* ticks in Florida [41], together with the recent genomic characterization of new LCMV strains in several tick species in northeastern China [42], suggests that ticks might contribute to mammarenavirus circulation in nature, which could have a significant impact on mammarenavirus epidemiology. Although the precise role ticks play in mammarenavirus biology and epidemiology is yet to be determined, the presence of the LCMV and TCRV in host-seeking ticks represented an important finding that motivated us to search for additional mammarenaviruses in these potential vectors. Using NGS, which has demonstrable power in unbiased pathogen discovery [78–80], we examined *A. americanum*-derived samples collected in Florida [41]. In addition to the already identified clade B NW mammarenavirus TCRV, we detected a replication-competent novel variant of the clade D TAMV, denominated TAMV-FL (Fig 1D). Both nucleotide and amino acid sequences of TAMV-FL differ significantly from TAMV-Ref and other known TAMV isolates [41,48]. However, despite the presence of replication-competent TAMV in host-seeking ticks, the virological and epidemiological role of arthropods in mammarenavirus biology is yet to be fully established and deserves further investigation. Moreover, even though being isolated in 1960s, TCRV-FL shows very few differences with the TCRV 11573 reference strain [81], while TAMV-FL significantly diverged at nucleotide and amino acid level, from previously isolated TAMV-Ref (W-10777) strain, indicating notable quasispecies plasticity.

Evaluation of potential zoonotic pathogens within natural reservoirs is of paramount importance for improving our ability to understand and control spillover events, as depicted by the current SARS-CoV-2 global pandemic [82]. Like other RNA viruses, mammarenaviruses are subjected to quasispecies dynamics, which facilitate the evolution of exceptional adaptive traits that may potentiate viral emergence [14,30,31]. Among clade B NW mammarenaviruses, the coincidental use of hTfR1 to mediate viral cell entry has been linked to the potential to cause severe human disease, thus hTfR1 usage could be viewed as an early indicator for potential emergence in NW mammarenavirus [21,27,28]. While clade D WWAV-Ref is not considered a human pathogen, the natural strain WWAV-AV96 is implicated in a small number of fatal human infections [20]. Importantly, similar to pathogenic clade B mammarenaviruses, WWAV-AV96 utilizes hTfR1 to enter human cells more efficiently than WWAV-Ref does, providing further evidence that hTfR1 usage may influence human disease potential also within clade D [25,51]. Nevertheless, in apparent contradiction to this trend, JUNV Candid#1 GP was found to bind hTfR1 more efficiently than its pathogenic XJ parental strain [83]. However, additional amino acid changes have been reported to contribute to attenuation of JUNV Candid#1 [84–86], suggestive that hTfR1 usage is not the sole parameter that dictates host tropism or virulence.

Our functional analyses revealed that reduced hTfR1 expression and treatment with an anti-hTfR1 blocking antibody reduce TAMV-FL infectivity, indicating the capacity of TAMV-FL to utilize hTfR1 for cell entry (Fig 2). Subsequent passaging of the tick-derived isolate in IFN-treated, human immunocompetent A549 cells allowed us to isolate TAMV-FL from co-existing TCRV (Fig 3) but also led to the generation and enrichment of two amino

acid substitutions, N151K and D156N, in GP1 of TAMV-FL in the viral quasispecies (Table 1). Notably, the presence of these two mutations was essentially mutually exclusive, with the double mutant being virtually absent in the observed viral populations. Although TAMV-Ref and TAMV-FL bind HSPG comparably (S5A and S5B Fig and S1 Text), the introduction of either the N151K or D156N substitution in the TAMV-FL backbone increased hTfR1 and heparan sulfate binding whilst reducing binding to ShTfR1 (Figs 5 and 6). Noteworthy, the double mutant harboring N151K/D156N substitutions displayed hTfR1 dependence comparable to the parental TAMV-FL but showed the highest binding capacity to heparan sulfate among all TAMV-FL variants. Our structure-based mapping reveals both N151K and D156 are likely to be structurally constrained in a location near the putative trimeric axis of mature GP, and distal to the predicted hTfR1-GP interaction interface (Fig 4). Furthermore, either of the single substitution also led to a delayed endosomal escape, from more acidified compartments (Fig 7). In the TAMV-Ref genetic backbone, however, either single substitution has similar effect regarding binding capacity to heparan sulfate, but does not cause delay in the endosomal escape (Fig 7).

Despite apparent host-specificity of clade A, B, and C NW mammarenaviruses, clade D NW mammarenaviruses can persist in different rodent species and diverse viruses can be isolated from the same species [15,75–80]. Such host species promiscuity suggests complementarity between the various GPs and TfR1 orthologues, which may facilitate cross-species transmission. Given the previously documented evolutionary receptor switch events [87,88], it is conceivable viral adaptation to new rodent hosts as well as to humans. The concurrence of increased binding capacity of TAMV-FL to both hTfR1 and ShTfR1 demands further investigation in order to elucidate the drivers for such adaptive ambivalence. Co-diversification of a mammarenavirus with its individual host (ticks, rodents or both) has resulted in GP architectures that are likely to be finely tuned for host-cell infection of the native reservoir and the usage of hTfR1 may be coincidental. It is of note that 3.8% of 131 Seminole Native American individuals sampled in Southern Florida were seropositive for TAMV [81]. This suggests at least sporadic zoonotic events involving TAMV in regions coincident with the geographic distribution of the TAMV reservoir, *S. hispidus* [81]. However, TAMV has yet to be isolated from human tissues or directly associated with human disease, suggesting that additional factors may determine efficient TAMV pathogenesis upon zoonotic spillover. Nevertheless, 1% of 1185 central nervous system-affected hospitalized cases were seropositive for WWAV, suggesting that clade D NW mammarenaviruses cause either severe undiagnosed disease or subclinical infections [20,81].

The hTfR1-tropic clade D NW mammarenavirus WWAV-AV96 GP1, harbors the substitution D154N (equivalent to D156N in TAMV-FL GP1) and exhibits high variability at GP1 position 149 (equivalent to position 151 in TAMV-FL GP1) when compared to other WWAV strains (Fig 4) [25,89]. Although previous genetic analysis of WWAV strains did not address the potential for quasispecies complexity within viral populations, the WWAV-AV96 consensus sequence indicates a major abundance of the D154N substitution [25]. The functionality of the WWAV-AV96 GP1 substitution S149G remains unknown. The presence of equivalent substitutions D156N and D154N in GP1 of TAMV-FLp5s and WWAV-AV96, respectively, further suggests the possibility for parallel evolution of both viruses and the existence of similar selective pressures, converging on similar abilities to use hTfR1 (Fig 5) or increased binding capacity for heparan sulfate [90] (Fig 6). Interestingly, the loss of hTfR1 dependence of TAMV-FL DM may explain the almost complete absence of the double mutant N151K/D156N in the TAMV-FLp5s and TAMV-FLp5l samples and the existence of D154N substitution in WWAV-AV96 but not in position 149. However, the multiple phenotypic changes associated with the TAMV-FL GP1 D156N substitution currently hampers discrimination of

the specific selectable trait. Despite a lack of phenotypic characterization of these mutations in TAMV's natural host, it is likely that the N151K and D156N variants only occur or increase their relative abundance during infection of human cells, as they would be likely subjected to negative selective pressure during infection in *S. hispidus*. Indeed, the low relative abundance of N151K (0.11%) and complete absence of D156N substitutions in the parental TAMV-FL population support this hypothesis. Future investigations into TAMV will focus on whether such substitutions are also observed during natural human infection.

Although there is no current evidence of the location of the initial replication sites upon endosomal escape, mammarenaviruses are known to induce perinuclear structures for viral replication and transcription [83]. Therefore delayed endosomal escape may facilitate genome delivery to this area and prevent innate immune defense by hiding viral RNA in endosomal compartments during transport. Interestingly, compared to other mammarenaviruses, the OW mammarenavirus LASV escapes from unusual acidic endosomal compartments [91–93], and this event occurs from less acidic compartments in absence of the intracellular receptor LAMP1 [91], suggesting a strong dependence on LAMP1 for efficient endosomal escape and a benefit from delayed endosomal escape. Nevertheless, although additional studies are encouraged, the available data suggest that delayed endosomal escape may constitute a selective advantage for mammarenaviruses. The location of N151K and D156N, close to the trimeric axis of GP (Fig 4), may provide a plausible explanation for the observed delayed endosomal escape (Fig 7), by affecting the overall GP stability and the pH-dependent membrane fusion. Moreover, our mapping analyses (Fig 4) suggest an unknown role for non-receptor-binding site GP1 residues in the modulation of TAMV receptor tropism, such as changes in the pre-fusion structural dynamics relevant to receptor binding, long-distance allosteric effects, or GP stability. Low GP1 sequence conservation among even closely related NW mammarenaviruses renders any correlation of GP1 sequence motifs with receptor specificity challenging. Indeed, in some instances are no obvious structural incompatibilities that would prevent hTfR1 binding of non-pathogenic NW mammarenaviruses [25]. Therefore, our discovery that residues 151 and 156 can modulate hTfR1 usage broadens our understanding of GP1 residues that contribute to receptor usage and tropism. We also reported first evidence of GP-induced syncytia formation at neutral pH, which is an exclusive trait of TAMV-Ref among mammarenaviruses, and it is increased by D156N substitution (S6 Fig and S2 Text). Nevertheless, the possible correlation of syncytia formation with the different architectures of TAMV-Ref and TAMV-FL GPs as well as its biological consequences remains unclear.

Adaptation towards increased HSPG affinity has been previously documented for other viruses because of tissue culture adaptation or natural evolution [35–40,68]. With limited exceptions, HSPG serve as attachment factors that increase local virus concentration at the cell surface, where the switch to a *bona fide* receptor may take place [39]. Interestingly, despite preferential HSPG binding conferred by the double substitution N151K/D156N (Fig 6), the detrimental effect on hTfR1 usage seems to favor the selection of single mutants (Fig 5). The pathogenic WWAV-AV96 exhibits increased HSPG binding compared to its closer relative WWAV-Ref, demonstrating that adaptation of clade D NW mammarenaviruses towards increased HSPG binding occurs both within the laboratory and in nature.

Although hTfR1 use is an important trait to evaluate the potential for breaking the species barrier, zoonotic events are complex multi-factorial processes. Despite the extensive characterization of TAMV GP-mediated entry performed in the present study, further investigation is required to elucidate the epidemiological consequences of the phenotypic changes described in TAMV-FL and the variants that occurred during *in vitro* passaging in human cells.

In sum, we have identified a novel circulating TAMV strain in ticks, TAMV-FL, which contrary to TAMV-Ref is capable of utilizing hTfR1 for cell entry. Furthermore, our findings

indicate that TAMV-FL quasispecies can rapidly acquire new adaptive traits, including increased hTfR1 usage, enhanced HSPG binding, and delayed endosomal escape. Although further investigation is required to address the epidemiological consequences of these findings, the observed phenotypic changes in TAMV-FL and its quasispecies variants, suggest potential for zoonotic spillover. The potential for hTfR1-mediated transmission into human populations, combined with the observed genetic plasticity and phenotypic changes *in vitro*, warrants increased efforts to monitor TAMV-FL persistence in animal and arthropod hosts.

## Materials and methods

### Antibodies, plasmids and reagents

Mouse monoclonal antibodies (mAb) MA03-BE06 and IC06-BE10 [65] were obtained from BEI Resources (Manassas, VA). Alexa Fluor 488 F(ab′)2 fragment of goat anti-mouse IgG and Alexa Fluor 594 goat anti-mouse IgG were purchased from Life Technologies (Carlsbad, CA). Anti hTfR1 (CD71, Ref#555534) was obtained from BD Biosciences and isotype control (Ref#11711) were purchased from R&D systems. Anti HA (High Affinity) rat monoclonal IgG1 antibody (Cat. No. 11 867 423 001) and recombinant human IFN (Interferon-αA/D human, Cat# I4401) were purchased from Sigma-Aldrich (St. Louis, Missouri, USA). Plasmid lentiCRISPR v2 was a gift from Feng Zhang (Addgene plasmid # 52961; http://n2t.net/addgene:52961). Lentiviral packaging plasmid pCMV-VSV-G was a gift from Bob Weinberg (Addgene plasmid # 8454; http://n2t.net/addgene:8454) and pLJM1-EGFP was a gift from David Sabatini (Addgene plasmid # 19319; http://n2t.net/addgene:19319). HA-tagged GPCs of TAMV-Ref (NC_010701.1), TAMV-FL (MK500937.1), and WWAV-AV96 (EU 123330.1) were synthetized *in vitro* (Genscript, Piscataway, NJ, USA). All TAMV GPs were cloned into pCAGGS plasmid, between BglII and XhoI restriction sites and HA-tagged. HA-tagged ShTfR1 was *in vitro* synthetized by Genscript (Piscataway, NJ, USA) and subcloned into pcDNA3.1(+) vector.

### Viruses and cells

Human lung adenocarcinoma epithelial cells (A549), human embryonic kidney cells (293T) and African green monkey kidney epithelial cells (VeroE6) were maintained in Dulbecco's modified Eagle medium containing high glucose (4.5 mg/l) and GlutaMAX (DMEM, Gibco BRL) supplemented with 10% (vol/vol) fetal calf serum (FCS) at 37˚C and 5% (vol/vol) $CO_2$. Baby hamster kidney cells (BHK)-21 were maintained in DMEM supplemented with 5% (vol/vol) FCS and non-essential amino acids (Gibco BRL) at 37˚C and 5% (vol/vol) $CO_2$. All cells were regularly tested for mycoplasma with MycoAlert Mycoplasma detection kit (Lonza). The tick-derived isolate was kindly provided by Dr. Katherine Sayler, and was the very same sample previously reported to contain TCRV [41]. Tick homogenates were used to infect VeroE6 cells, and passaged 3 times. Supernatants were cleared by brief centrifugation (1500 rpm, 4˚C, 5 min) [41]. Viral stocks were produced by infecting BHK-21 cells with passage 3 (on VeroE6 cells) of tick-derived isolate, collecting the conditioned TCS after 4 days after infection and cleared by centrifugation at 1500 rpm for 5 minutes at 4˚C. Virus stocks were mixed 1:1 with a sterile solution of 140 g/l polyethylene glycol (PEG)-8000 in PBS and incubated overnight in rocking station at 4˚C. Samples were centrifuged at 8000 x g for 1h at 4˚C, TCS were discarded and precipitated material was resuspended in supplemented DMEM. PEG-precipitated samples were then layered on 30% (wt/vol) sucrose cushion and centrifuged at 37900 x g for 2 h at 4˚C in an Optima XPN-80 ultracentrifuge (Beckman Coulter) equipped with a SW-55 Ti rotor. After ultracentrifugation, pellets were resuspended in complete DMEM and stored at -80˚C.

## RNA extraction and library preparation

For RNA extraction, samples were layered on 30% (wt/vol) sucrose cushion and centrifuged at 100000 x g for 2 h at 4˚C in an Optima XPN-80 ultracentrifuge (Beckman Coulter) with a SW-55 Ti rotor. After ultracentrifugation, pellets were resuspended in nucleases buffer (Tris-HCl, pH 7.5 40mM, MgCl$_2$ 6mM, NaCl 10mM, CaCl$_2$ 1mM) and treated for 30 minutes at room temperature with nucleases cocktail to remove non-encapsided nucleic acids. Nucleases cocktail included DNase I (Catalog number EN0521) and RNAse A (Catalog number EN0531A) from Thermo Fisher Scientific (Waltham, Massachusetts, USA) and Nuclease S7 from Sigma-Aldrich (St. Louis, Missouri, USA) (Catalog number N5386). Reaction was stopped by addition of EDTA (28 mM) and EGTA (2 mM). RNA was then extracted with TRIzol reagent (Ambion) followed by ethanol precipitation and resuspended in 20 μl of nuclease-free water. cDNA libraries were prepared with TruSeq Stranded mRNA LT Sample Prep Kit or Kit and following manufacturer's instructions. Briefly, RNA was fragmented and first and second cDNA strands were synthetized. Then, 3' ends of cDNA were adenylated and adaptors were incorporated. Samples were then run in a MiSeq instrument (2x250 reads) (Illumina).

## NGS analysis

Raw reads were first quality filtered using PRINSEQ v0.20.4 [94]. A minimum average quality of 25 and a minimum length of 120 nt were required. Only paired reads were considered for further analysis. The taxonomy binning of the reads was carried out using Kraken v1.0 [95] using the pre-built DustMasked MiniKraken DB 8GB (built on 18/10/17). Spades v3.12.0 [96] was used for contig assembly (using—careful option). For contig classification, contigs were aligned using Blastn against the reference viral genomes NCBI database (downloaded on 07/01/19), and only those hits of at least 500 bp with and e-value $< 1x10^{-5}$ were considered. Quality filtered reads were aligned against reference viral genomes (TCRV and TAMV) o against new sequence genomes/contigs using Bowtie2 v2.3.4.3 [97] using "local" mode. Alignment pileups were built using Samtools v 0.1.19 [98] (minimum required quality 20). Then, pileups were used for SNVs detected using VarScan v2.4.0 [99] (minimum average quality 20, minimum coverage 20 and strand-filtering 80). Sequence conservation between TAMV strains was calculated using mVista program (http://genome.lbl.gov/vista/mvista). Raw reads where submitted to European Nucleotide Archive (ENA) under project accession PRJEB31100.

## Pseudotyped virus production

The pseudotype viral system was based on the recombinant VSVΔG-EGFP/Luc (Indiana strain) vector in which the glycoprotein gene (G) had been deleted and replaced with genes encoding green fluorescent protein (EGFP) and luciferase (Luc) [100]. To produce VSV-based pseudoviruses (PV) decorated with viral GP of interest, we performed as described in [101]. Briefly, 293T cells were plated in 10cm dishes and transfected with plasmids encoding for selected viral glycoproteins using JetPrime transfection kit (PolyPlus) and followed manufacturer's instructions. After 36 hours post transfection, cells were infected with VSVΔG-EGFP/Luc at MOI of 3 PFU/cell and incubated at 37˚C and 5% (vol/vol) CO$_2$ for 90 minutes. After incubation, fresh supplemented DMEM was added with neutralizing anti-VSV monoclonal antibody I-1 (conditioned hybridoma supernatant diluted 1:100). After 16h post infection, conditioned TCS were collected, cleared by centrifugation at 1500 rpm for 5 minutes at 4˚C, mixed 1:1 with a sterile solution of 140 g/l PEG-8000 in PBS, and incubated overnight in rocking station. Samples were centrifuged at 8000 x g for 1h at 4˚C. TCS were discarded and pellets were resuspended in supplemented DMEM.

## Immunoblotting

For GP and VSV-M quantitation, PV preparations were cleared by brief centrifugation (1500 rpm, 5 min, 4°C) and purified by ultracentrifugation in 30% sucrose cushion (in PBS). Pellets were re-suspended in DMEM, and a fraction was lysed with Laemli buffer. Then, proteins were separated by SDS-PAGE and transferred to nitrocellulose membrane. After blocking in 5% (wt/vol) skim milk in PBS, 0.1% (vol/vol) Tween-20 (PBST), membranes were incubated with primary antibody (Anti VSV-M monoclonal antibody from Merck, reference MABF2347, and anti HA high affinity antibody from Merck, reference 11867423001) in 5% (wt/vol) skim milk, PBST, for 2h at room temperature. Then, membranes were washed 3 times with PBST, and HRP-coupled secondary antibodies were applied 1:5000 in PBST (1 h, room temperature). Membranes were then washed with PBST, blots were developed by enhanced chemiluminescence (ECL) using LiteABlot kit (EuroClone). Signals were acquired by ImageQuant LAS 4000Mini (GE Healthcare Lifesciences). Signals were acquired with an ImageQuant LAS 4000Mini (GE Healthcare Lifesciences, Glattbrugg, Switzerland) instrument and quantified with ImageJ Software.

## Virus and pseudovirus infections and titrations

Virus infections were performed as described in [64]. Briefly, viral infections were performed by removing cell TCS, adding viral inoculum and incubation during 90 minutes at 37°C and 5% (vol/vol) $CO_2$. Inoculums were then removed and fresh supplemented DMEM was added. For viral titrations, samples were 10-fold serially diluted in supplemented DMEM and used to infect VeroE6 cells (96 well-format plates). After 16-20h, cells were washed once with PBS, fixed with 2% formaldehyde/PBS and stained for viral NP with MA03 or IC06 antibodies. Positive infectious foci were scored using an EVOS Floid Cell Imaging Station 20X Plan fluorite lens (Thermo Fisher Scientific, Waltham, Massachusetts, USA). PV infections were performed adding 150–300 PFU/well (96 well-format plate) diluted in supplemented DMEM and used as inoculum. Cells and inoculum were incubated for 90 minutes in 37°C and 5% (vol/vol) $CO_2$. Then, TCS were removed and fresh supplemented DMEM was added to each well. After 16–20 hours, cells were washed once with PBS and fixed with 2% formaldehyde/PBS for EGFP-positive scoring or assayed for luciferase activity. Positive infectious foci were scored using an EVOS Floid Cell Imaging Station 20X Plan fluorite lens (Thermo Fisher Scientific, Waltham, Massachusetts, USA) or luciferase activity measured by ONE-Glo Luciferase Assay System, from Promega (Madison, Wisconsin, USA), as described by the manufacturer.

## Antibody perturbation assays

A549 or 293T cells were plated in 96 well-plate format 24h before infection. Cells were preincubated with 200 nM of anti-hTfR1 or isotype control antibodies for 1h at RT. Cells were then infected with 150–300 PV PFU in presence of indicated antibodies. After adsorption, TCS were removed and 20mM $NH_4Cl$ in supplemented DMEM was added to each well. After 16–20 hours, cells were washed once with PBS and fixed with 2% formaldehyde/PBS for EGFP-positive scoring or assayed for luciferase activity as described above.

## Fluorescence-activated cell sorting (FACS)

Cells were detached, resuspended, and washed once with supplemented DMEM, washed once with PBS, and fixed with 2% (wt/vol) formaldehyde in PBS for 30 minutes at room temperature. After washing with PBS, cells were permeabilized (for MA03 and IC06 antibodies) or left un-permeabilized (for anti-hTfR1 or anti HA antibodies) for 30 minutes at room temperature

with 1% (vol/vol) FCS in PBS with 0.1% (wt/vol) saponin (permeabilization solution) or with-out (surface staining solution). Primary and secondary antibodies were diluted in permeabili-zation or surface staining solution and incubated for 1h and 45 minutes, respectively, at room temperature. Cells were washed three times with PBS and analyzed with a FACS Calibur flow cytometer (Becton Dickinson, San Jose, CA).

## Immunofocus assay (IFA)

Infected samples were washed once with PBS and fixed with 2% (wt/vol) formaldehyde in PBS for 30 minutes at room temperature. Cells were washed with PBS and permeabilized for 30 min-utes at room temperature with 0.1% (wt/vol) saponin and 1% (vol/vol) FCS in PBS (permeabili-zation solution). Primary and secondary antibodies were diluted in permeabilization solution and incubated for 60 and 45 minutes, respectively, at room temperature. Cells were washed three times with PBS. Positive infectious foci were scored using an EVOS Floid Cell Imaging Station 20X Plan fluorite lens (Thermo Fisher Scientific, Waltham, Massachusetts, USA).

## Endosomal escape and ammonium chloride titration

To monitor endosomal escape kinetics, we performed as described in [102]. Briefly, to ensure synchronized infections, A549 cells were cooled down to 4˚C and infected with pre-cooled PVs as described above. To allow attachment but not internalization, cells were incubated on ice for 90 minutes and then quickly shifted to 37˚C. Upon infection, equal volume of ammo-nium chloride 40 mM in supplemented DMEM was added to each well (20mM final concen-tration) at indicated time points. After 16–20 hours, cells were washed once with PBS and fixed with 2% formaldehyde/PBS for EGFP-positive scoring. Relative entry was referenced to values obtained 30 minutes after infection. To determine ammonium chloride sensitivity, PV infections were made in A549 cells. Cells were pre-treated with indicated concentrations of ammonium chloride for 30 minutes before infection. Cells were PV infected as described above, in presence of indicated concentrations of ammonium chloride. After inoculum removal, fresh supplemented DMEM with indicated concentrations of ammonium chloride were added. After 16–20 hours, cells were washed once with PBS and fixed with 2% formalde-hyde/PBS for EGFP-positive scoring. Relative entry was referenced to infectivity in absence of ammonium chloride.

## Structure-based mapping analysis

Amino acid substitutions identified within the TAMV quasispecies were mapped onto the crystal structure of the GP1 glycoprotein from WWAV (PDB: 6HJ5) using PyMOL (The PyMOL Molecular Graphics System, Version 2.0 Schrödinger, LLC), based upon positions inferred from amino acid sequence alignment calculated using MultAlin [103]. A model for the putative trimeric arrangement of NW GP1s, representative of a mature mammarenavirus GP ectodomain, was generated by superposition of WWAV GP1 onto the structure of the LASV GP ectodomain (PDB: 5VK2), using secondary-structure matching superposition [104] implemented in COOT [105]. The putative TfR1 binding interface on the clade D GP1 surface was predicted using PDBePISA [106], from a model constructed by superimposing WWAV GP1 onto the structure of MACV GP1 in complex with hTfR1 (PDB: 3KAS).

## CRISPR/Cas9

A549 cells express hTfR1, which mediates entry of hTfR1-trofic mammarenaviruses [28]. To address the changes in viral entry, we aimed to deplete hTfR1 from A549 cells using CRSIPR/

Cas9 editing. To reduce hTfR1 expression, A549 cells were subjected to CRISPR/Cas9 engineering as described in [107,108]. Briefly, single guide RNA (sgRNA) sequences targeting the exon regions of hTfR1 (5′- ATCACTATAGATCCATTCAC-3′). Annealed oligonucleotides were cloned into pLenti CRISPRv2 ccdB by digesting oligonucleotides and vector with BsmBI. In order to transduce A549 cells with the respective gRNA sequence, lentiviral VSV-G pseudotyped particles were produced in the cells. 293T cells were then transfected with plasmid DNA encoding the human immune deficiency virus (HIV) Gag and Pol proteins, VSV-G protein, and the respective pLenti CRISPR construct. Lentiviruses were harvested 48 h after transfection, and A549 cells were transduced with the lentiviruses for 8 h. 37˚C and 5% (vol/vol) $CO_2$. After 48 h after transduction, cells were subjected to puromycin selection (2 μg/ml) for 14 days. Upon puromycin selection, cells were sorted with a Beckman Coulter MoFlo Astrios EQ instrument, discarding those cells that showed higher fluorescence than the isotype control-stained population. Samples were routinely characterized for hTfR1 expression by FACS for every experiment performed.

## siRNA transient gene knockdown

To address the impact of transient reduced expression of hTfR1, we used 293T cells, which express hTfR1 and are susceptible to high-efficiency transfection. siRNA TFRC (7037) RNA probe was obtained from Dharmacon/Horizon Discovery (Cambridge, UK). 293T cells were transfected with siRNA probe with RNAiMAX transfection reagent (Thermo scientific, Waltham, Massachusetts, USA), following manufacturer's instructions. 24 hours upon transfection, cells were plated in poly-lysine-treated 96 plates. 24 hours after plating, cells were infected with PV and tested for hTfR1 expression. After 16–20 hours, cells were washed once with PBS and fixed with 2% formaldehyde/PBS for EGFP-positive scoring or assayed for luciferase activity.

## Heparin inhibition and heparinase III treatment

In order to evaluate the binding capacity of selected viral GP, PV infections were performed in presence of heparin or in heparinase III-treated A549 cells. For infections in presence of heparin, PV were diluted in supplemented DMEM with indicated heparin concentrations (Catalog number H9267. Sigma-Aldrich, St. Louis, Missouri, USA) for 1 hour at 4˚C. Upon incubation, TCS was removed and PV infections were performed as described above. To remove heparan sulfate proteoglycans from A549 cell surface we performed as described in [69]. Briefly, we plated A549 cells in 96 well-plate format 24 hours before infection. TCS were removed and heparinase III buffer (NaCl 100 mM, Tris-HCl 20mM and $CaCl_2$ 1.5mM) with 10 IU heparinase III (Catalog number P0737S. New England Biolabs, Ipswich, Massachusetts, USA) was added and incubated for 2 hours at 30˚C and 5% (vol/vol) $CO_2$. After heparinase III treatment, cells were washed there times with PBS and PV infections were performed as described above.

## Statistical analysis

Data analyses were performed with Excel software using ANOVA test (Significance 0.05 if not otherwise stated). The half-escape time and $IC_{50}$ were calculated using Prism software version 8.2.1 (GraphPad Software, San Diego, CA, USA, Version 8.2.1).

## Supporting information

**S1 Text. TAMV-Ref and TAMV-FL similarly bind heparan sulfate proteoglycans.** (DOCX)

**S2 Text. TAMV-Ref GP but not TAMV-FL GP induce syncytia formation at neutral pH, and it is increased by D156N mutation.**
(DOCX)

**S1 Fig. Alignment of TAMV available sequences.** For S segment, TAMV-Ref sequence [47], TAMV W10777 [49], TAMV-FL (MK500936) and TAMV AV97140103 (Abbreviated as TAMV-AV97; EU486821.1) were used. For L segment TAMV-Ref sequence [47], TAMV W10777 [49], and TAMV-FL (MK500937). SNVs are highlighted with yellow boxes, and bold letters are used for the most frequent base at each position.
(TIF)

**S2 Fig. GP incorporation into PV. (A)** TAMV-Ref and TAMV-FL PV were purified by ultra-centrifugation in 30% sucrose cushion (3h, 100,000 x g, 4°C), lysed, separated by SDS-PAGE and assayed for TAMV-FL GP2 detection (anti-HA monoclonal antibody) and $VSV_M$ (specific VSV-M antibody) in immunoblotting. **(B)** TAMV-FL PV were produced by transfecting indicated amounts of $TAMV-FL_{GP}$ DNA. Obtained PV preparations were purified by ultracentrifugation in 30% sucrose cushion (3h, 100000 x g, 4°C), lysed, separated by SDS-PAGE and assayed for TAMV-FL GP2 detection (anti-HA monoclonal antibody) by immunoblotting. **(C)** A549 cells were infected with equal volumes of TAMV-FL preparations produced in (B). **(D)** A549 cells were infected with amounts of TAMV-FL normalized with obtained GP-signal in (B). Error bars in panels (C) and (D) represent standard deviations (n = 3). Asterisks in panels (C) and (D) denote statistical significance in ANOVA test (ns: $p > 0.05$; *: $p < 0.05$).
(TIF)

**S3 Fig. Transferrin receptor 1 expression. (A)** Endogenous hTfR1 expression in siRNA hTfR1-transfected 293T cells. Non-targeted probe (scrambled siRNA) was used as control. Receptor expression was assessed by specific hTfR1 (CD71) monoclonal antibody or respective isotype control. **(B)** Endogenous hTfR1 expression in A549/hTfR1 KD cells obtained by CRISPR/Cas9. Non-transduced parental A549 cells were used as control. Receptor expression was assessed by specific hTfR1 (CD71) monoclonal antibody or respective isotype control. Endogenous hTfR1 expression in **(C)** 293T and **(D)** A549 cells. Respective isotype control was used as negative control. **(E)** Ectopic expression of *Sigmodon hispidus* (Sh)TfR1 expression in 293T cells. 293T cells were transfected with HA-tagged plasmid for transient ShTfR1 expression. Receptor expression was monitored with specific anti HA monoclonal antibody at the same time than infections were carried out.
(TIF)

**S4 Fig. GP incorporation into TAMV-FL and TAMV-FL mutants PVs.** TAMV-FL and TAMV-FL mutant PVs were purified by ultracentrifugation in 30% sucrose cushion (3h, 100000 x g, 4°C), lysed, separated by SDS-PAGE and assayed for TAMV-FL GP2 detection (anti-HA monoclonal antibody) and $VSV_M$ (specific VSV-M antibody) in immunoblotting.
(TIF)

**S5 Fig. Heparan sulfate proteoglycans dependence of TAMV-Ref and TAMV-FL. (A)** Relative entry in infections performed in A549 cells in presence of increased concentrations of heparin. Error bars represent standard deviations (n = 4). **(B)** Infections performed in heparinase III-A549-treated cells. Error bars represent standard deviations (n = 4). Asterisks in all panels denote statistical significance in ANOVA test (ns: $p > 0.05$; *: $p < 0.05$; **: $p < 0.01$).
(TIF)

**S6 Fig. Syncytia formation of TAMV-FL and TAMV-FL mutants and WWAV strains.**
**(A)** Scheme of syncytia formation monitoring by co-culture of co-transfected 293T cells. LgBit

LUC and HiBit LUC protein fragments are only functional when co-expressed in the same cell. Syncytia formation upon clade D NW arenavirus GP transfection in 293T cells monitored by **(B)** luciferase activity (error bars represent standard deviations of n = 4) or **(C)** under fluorescence microscope (Scale bars represent 100 μm).
(TIF)

**S1 Table. Taxonomical reads classification obtained from tick-derived sample (done with Kraken) in the NGS run.**
(DOCX)

**S2 Table. Taxonomical contig classification obtained from tick-derived sample (done with Blastn) in the NGS run.**
(DOCX)

**S3 Table. Sequence comparison of TCRV-FL, TCRV-11573, TCRV-Florida and TCRV-BEI.**
(DOCX)

## Acknowledgments

Sadly, our colleague and friend Stefan Kunz passed away on 10th January 2020. The authors want to dedicate this study to his memory. The authors thank Dr. Katherine Sayler (Department of Large Animal Clinical Sciences, College of Veterinary Medicine, University of Florida) for the tick-derived isolate and Dr. Paula M Cannon for providing the plasmid encoding for WWAV-Ref GP. We would like to thank Prof. Juan Carlos de la Torre and Dr. Jérôme Gouttenoire for enriching scientific discussion and advice. We are truly thankful to Dr. Roberto Balbontin for his scientific feedback, critical thinking and discussion as well as for his help in the manuscript refinement.

## Author Contributions

**Conceptualization:** Hector Moreno, Stefan Kunz.

**Formal analysis:** Hector Moreno, Alberto Rastrojo, Rhys Pryce, Thomas A. Bowden.

**Funding acquisition:** Stefan Kunz.

**Investigation:** Hector Moreno.

**Methodology:** Hector Moreno, Alberto Rastrojo, Rhys Pryce.

**Project administration:** Hector Moreno.

**Resources:** Gert Zimmer, Gisa Gerold.

**Supervision:** Hector Moreno.

**Visualization:** Hector Moreno, Alberto Rastrojo, Rhys Pryce.

**Writing – original draft:** Hector Moreno, Stefan Kunz.

**Writing – review & editing:** Hector Moreno, Alberto Rastrojo, Rhys Pryce, Chiara Fedeli, Gert Zimmer, Thomas A. Bowden, Gisa Gerold.

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
