## [Decision Letter · Decision Letter 0]

18 Aug 2020

Dear Dr. Moreno,

Thank you very much for submitting your manuscript "A NOVEL CIRCULATING TAMIAMI MAMMARENAVIRUS SHOWS POTENTIAL FOR ZOONOTIC SPILLOVER" for consideration at PLOS Neglected Tropical Diseases. As with all papers reviewed by the journal, your manuscript was reviewed by members of the editorial board and by several independent reviewers. In light of the reviews (below this email), we would like to invite the resubmission of a significantly-revised version that takes into account the reviewers' comments. 

All three reviewers express interest in the study, but one reviewer have several criticisms and suggestions, which need to be addressed.

We cannot make any decision about publication until we have seen the revised manuscript and your response to the reviewers' comments. Your revised manuscript is also likely to be sent to reviewers for further evaluation.

Sincerely,

Jonas Klingström

Associate Editor

Jeremy Camp

Deputy Editor

Reviewer's Responses to Questions

**Key Review Criteria Required for Acceptance?**

**Methods**

-Are the objectives of the study clearly articulated with a clear testable hypothesis stated?

-Is the study design appropriate to address the stated objectives?

-Is the population clearly described and appropriate for the hypothesis being tested?

-Is the sample size sufficient to ensure adequate power to address the hypothesis being tested?

-Were correct statistical analysis used to support conclusions?

-Are there concerns about ethical or regulatory requirements being met?

Reviewer #1: The methods used are appropriate, as well as the data analysis.

Reviewer #2: 1) Despite the request of the previous reviewers, statistical assessment is still missing for some experiments (e.g. Figs 2E, 6 and 7, S2 and S3B). Further, many of the figures fails to indicate what statistical test was performed and this information also appears to be missing in the methods. Where the test is indicated, the authors say they used a T-test, which is inappropriate for the analysis of datasets containing more than 2 samples (even where only pair-wise comparisons of samples are subsequently analysed). Rather ANOVA needs to used.

2) Lines 206-208. Here (and also elsewhere in the text) the authors appear to be trying to use the strength of their p values as an indication of the strength of their biological effect. This is not what p-values indicate and they simply cannot be used in this way.

3) In assessing TAMV strain diversity as it relates to their new TAMV-FL isolate the authors need to include values for the comparison to the much more recently isolated TAMV sequence for strain CDC W-10777 from 1997, which is publically available (e.g. In GenBank). 

4) The authors need to clearly indicate the provenance of their sample in the methods. This is vaguely alluded to in the acknowledgements and the use of references, but it needs to be made clear if this is the exact same tick pool that Sayler et al. used in their publication of a newly identified TCRV sequence. If this is the case, it is not surprising that the authors observe a TCRV sequence with similarly high sequence identity to that previously reported, whereas if this is a different tick pool this observation gains increased significance. 

In either case, it is important to note that the choice of reference sequence used by the authors for their analysis of the TCRV sequence is not suitable, as this sequence has been repeatedly shown to contain major discrepancies compared to all other sequences of the same isolate (e.g. PMID: 26587982 (accession: KP159416 ), PMID: 23382389 (accession: KC329849)) and that these account for almost all of the difference between the Florida and the original TRVL-11573 isolates (PMID: 32462284, accession numbers: MT081316, MT081317). Any analysis of their identified TCRV sequences should use these newer sequences for comparison.

Reviewer #3: The methods are appropriate. Experiments have been conducted according to high standards and have been controlled via appropriate controls.

**Results**

-Does the analysis presented match the analysis plan?

-Are the results clearly and completely presented?

-Are the figures (Tables, Images) of sufficient quality for clarity?

Reviewer #1: The results are adequately presented.

Reviewer #2: 1) I can appreciate that the requests of the previous reviewers to obtain a non-adapted TAMV-FL for confirmatory work to support the authors’ findings with VSV pseudotypes, may go beyond the scope of the present study. However, if the authors are going to depend exclusively on these pseudotype data for some of their work, they need at least to confirm that their VSV pseudotypes contain equivalent amounts of GP incorporation. Indeed, could such differences possibly explain for example the striking difference in the role of hTfR1 in WWAV when using antibody vs KD cells (Fig 5A vs 5B)? This leads the authors to strongly assert that WWAV-Ref does not use hTfR1 (Line 455-458), although this is actually contradicted by some of their own data (Figure 5B).

Further, if the authors are unable to do confirmatory work with actual virus (which is justifiably the case here) they still need to directly acknowledge this limitation of the study.

2) Based on what the authors write in the methods it appears that they did not synchronize their infections for the ammonium chloride escape assays. This is clearly needed in this case since the authors show there are indeed differences in binding between these pseudotypes. Otherwise, the influence on binding may convolute the authors findings, which involve changes in escape time within a very short window. Indeed, might this help explain some of the internal inconsistencies observed in the authors data between the ammonuium chloride concentration and time of escape data? e.g. TAMV-REF and JUNV have the same IC50 but very different escape times, while JUNV and TAMV-FL have the same escape times, but quite different IC50s. Similarly, WWAV-REF and WWAV-AV96, differ substantially in IC50 but have almost identical escape times. The authors also need to clearly address/explain these discrepancies, as they otherwise might call into question the reliability of this assay and/or the biological relevance of differences seen on this time scale.

3) Why do the authors only look at heparin usage but not hTfR1 dependence using their mutants (N151K, D156N and DM) in the TAMV-REF background?

Reviewer #3: Results are clearly presented in Figures, Tables and text. Figures and Tables are of high quality.

**Conclusions**

-Are the conclusions supported by the data presented?

-Are the limitations of analysis clearly described?

-Do the authors discuss how these data can be helpful to advance our understanding of the topic under study?

-Is public health relevance addressed?

Reviewer #1: The conclusions are fully supported by the results.

Reviewer #2: 1) As noted above (and also identified as an issue in the previous revision of the paper for PLoS Pathogens), the authors’ statements appear to misleadingly suggest that the identification of virus in blood-feeding arthropods can be used to presume an epidemiologically relevant function as a vector (e.g. Lines 37-40, Line 110-112, Line 435-439). In the absence of data for efficient virus replication, dissemination with the vector and effective transmission from the vector under relevant biological conditions these statements need to significantly moderated.

2) Also the authors’ assertion that their TAMV-FL isolate has “evolved to hTfR1 usage” is not necessarily supported by the data. Is it not just as likely (if not more so) that the reference strain of TAMV has evolved to have decreased dependence of hTfR1 as a result of its passage history over the last 50+ years? In that respect the authors also need to provide whatever information they have regarding the passage history of the TAMV isolate from which their TAMV-REF GP sequence is derived.

3) The authors refer to the “striking genetic plasticity” of TAMV (Lines 51-53), however, there is no indication as to what they are comparing to in order to make this assessment? Is the observed degree of variation so unusual for an arenavirus, either with respect to the extent of the divergence seen between the natural TAMV isolates (based on their respective dates of isolation), or in the number of changes seen following their 5 passages when transitioning to a cell culture based on a cell type derived from a different species? A concrete and objective basis for this assessment needs to be provided.

4) Work with the Candid#1 vaccine strain of JUNV shows that it has also acquired increased dependence on hTfR1 compared to virulent JUNV, and here it is posited to be associated with attenuation (PMID: 21976641). Thus it is not clear why the authors expect that in TAMV-FL a similar change due to passaging in human cells would constitute a predilection for adaptation to pathogenicity in humans, as they appear to be suggesting.

5) Line 521-524. The authors’ speculation that the observed delay in endosomal escape facilitates immune evasion during transport to the areas in which inclusion bodies occur is highly speculative. Firstly, this model is based on the supposition that genomes have to be transported to the sites of eventual inclusion body formation, which is not necessarily the case as these structures are the product of viral protein accumulation once genome delivery and viral RNA synthesis has already been successfully established. There is no evidence for instance that primary transcription takes place at these sites. Perhaps more importantly though, it overlooks other work on endosomal entry factors for arenaviruses suggesting that they act to enhance infection by facilitating early escape of the virus at higher pH thereby avoiding particle damage at low pH (PMID: 29295909). As such, it is not at all clear that delayed escape would be “a good thing” for the virus. Indeed, its harmful effects appear to be supported to some extent by the authors own observations that their double mutant is strongly disfavoured in nature, suggesting low viability. 

6) How do the authors suggest that the D156N mutant can show enhanced fusion at neutral pH (Figure S3) and yet show, if anything, slightly delayed fusion triggering (i.e. a need for lower pH for fusion) in the ammonium chloride escape assay (Fig 4)?

Reviewer #3: All conclusions are balanced and supported by the data. Public health relevance of the data is addressed in the Discussion.

**Editorial and Data Presentation Modifications?**

Reviewer #1: I recommend accepting the manuscript, as is.

Reviewer #2: 1) The authors are encouraged to also submit their TAMV-FL sequences to GenBank and include the corresponding accession numbers in the manuscript text.

2) Line 240-241. The authors should clarify for the reader that passage 2 in the long passage and passage 6 in the short passage amount to the same total amount of time for virus growth – which is also what one might expect.

3) “Frags” and “fragments” in Fig 1E should more correctly be ‘segments”.

4) Lines 442-445 and 459-461 are redundant.

5) Lines 469-472 is redundant with respect to the statements about heparin sulfate binding

Reviewer #3: (No Response)

**Summary and General Comments**

Reviewer #1: The manuscript by Moreno H. et al. describes the isolation of a new TAMV strain from ticks and the characterization of two amino-acid substitutions that promote hTfR1 and HSPG utilization, affect the spike triggering, as well as the cell penetration dynamics of the virus. The quick adaptation of the virus to hTfR1 suggests a potential capacity for spillover.

This is a highly timely and important manuscript in the arenaviral field. The results are exciting and will certainly contribute to the field. 

The manuscript has already gone a thorough round of revision, where the authors addressed comments made by three reviewers. 

I found the methods, display of results, and discussion to be appropriate. 

Except for a few occasional typos (i.g. line 407 "fig. 9c" should probably be "fig. 7c", for example), the manuscript is of high quality, and I thus recommend accepting it as is.

Reviewer #2: The manuscript by Moreno et al. describes the authors’ NGS sequencing of a new Tamiami virus (TAMV) isolate obtained from tick pools collected in Florida that have been previously reported to contain another arenavirus, Tacaribe virus (TCRV). The TAMV sequence identified is shown to be highly divergent from the reference strain from 1965, and thus may in its own right quite interesting with respect to understanding the extent of genetic variation of this virus. However, to fully appreciate this point the authors need to also include detailed information on comparisons of their sequence to the much more recently isolated TAMV sequences (strain CDC W-10777) from 1997, which are also publically available (e.g. In GenBank). 

Based on this sequence the authors go on to generate pseudotyped VSVs containing TAMV-FL, in order to compare usage of the human transferrin receptor and heparin to facilitate psuedotype particle entry. Further, they then use a quite clever approach (based on non-reciprocal superinfection exclusion) to isolate a passaged version of this new TAMV strain. During this process, the authors observe several mutation that become enriched during passage compared to their starting sequence and evaluate the relevance of these mutations for entry, including both receptor usage and the pH dependence of fusion. While these experiments are mostly well done, there a few points of concern that need to be clearly addressed. Specifically, the ammonium chloride escape experiments need to use synchronized infections, and the authors need to confirm that their pseudotypes have comparable (mature) GP incorporation. indeed these points may explain some of issues with internal consistency within the authors data, which they do not address.

Currently, the major weakness of the paper is that the authors make some broad provocative claims (including in the title) that are based on lines of argumentation that are not well developed and/or outright exceed what can reasonably be concluded based on the data.

Reviewer #3: The manuscript presented by Moreno et al. is an important addition to our understanding of adaptation processes that could facilitate an effective spread of arenaviruses from their natural reservoir to humans. The data are original and the experiments were carefully conducted and interpreted. In addition, the authors have responded to all comments of the previous reviewers and have thoroughly revised the manuscript. In my opinion, there is no need for further changes.

PLOS authors have the option to publish the peer review history of their article (what does this mean?). If published, this will include your full peer review and any attached files.

Reviewer #1: No

Reviewer #2: No

Reviewer #3: No
---

## [Decision Letter · Decision Letter 1]

13 Nov 2020

Dear Dr. Moreno,

Thank you very much for submitting your manuscript "A NOVEL CIRCULATING TAMIAMI MAMMARENAVIRUS SHOWS POTENTIAL FOR ZOONOTIC SPILLOVER" for consideration at PLOS Neglected Tropical Diseases. As with all papers reviewed by the journal, your manuscript was reviewed by members of the editorial board and by several independent reviewers. The reviewers appreciated the attention to an important topic. Based on the reviews, we are likely to accept this manuscript for publication, providing that you modify the manuscript according to the review recommendations. 

Sincerely,

Jonas Klingström

Associate Editor

Jeremy Camp

Deputy Editor

Reviewer's Responses to Questions

**Key Review Criteria Required for Acceptance?**

**Methods**

-Are the objectives of the study clearly articulated with a clear testable hypothesis stated?

-Is the study design appropriate to address the stated objectives?

-Is the population clearly described and appropriate for the hypothesis being tested?

-Is the sample size sufficient to ensure adequate power to address the hypothesis being tested?

-Were correct statistical analysis used to support conclusions?

-Are there concerns about ethical or regulatory requirements being met?

Reviewer #1: -

Reviewer #2: My comments have all been addressed.

Reviewer #3: The authors have appropriately responded to my comments on the original submission.

**Results**

-Does the analysis presented match the analysis plan?

-Are the results clearly and completely presented?

-Are the figures (Tables, Images) of sufficient quality for clarity?

Reviewer #1: -

Reviewer #2: My previous comments have been addressed. I have only a couple of minor additional comments on the results based on the revised version of the manuscript:

1) Fig S6B. The scale used here appears to be inappropriate for the data. It is impossible to estimate the increase between the very low values (below 1x107), while it is also necessary to break the scale at the top end to show the values for TAMV-REF D156N. This seems to be a clear indication that a log scale is needed here.

2) Table 1 and S3. I do not believe the term “single nucleotide variant” is appropriate here as this refers to minority populations present within a quasispecies, whereas what the authors are analyzing here are consensus sequences for different species and/or isolates of a virus – as such it would be more appropriate to refer to these simply as “nucleotide changes”. Also it is confusing to show the nucleotide comparison data twice (on the top and bottom of the matrix) and may even be unintentionally misleading to some readers (since amino acid comparisons are usually shown on the bottom of such matrices). Fields that are not needed should simply be blacked out.

Reviewer #3: The authors have appropriately responded to my comments on the original submission.

**Conclusions**

-Are the conclusions supported by the data presented?

-Are the limitations of analysis clearly described?

-Do the authors discuss how these data can be helpful to advance our understanding of the topic under study?

-Is public health relevance addressed?

Reviewer #1: -

Reviewer #2: My previous comments have been addressed well. However, based on changes to the text, there a couple of additional points identified where the discussion may benefit from minor modification:

1) Lines 540-544. While it is true that arenaviruses are generally isolated primarily from a specific species, there is also evidence for evolutionary host switching among rodent species. Do the authors believe that mechanisms analogous to what they see here could also be involved in facilitating that process?

2) Lines 513-6. The authors suggest that a recently reported V64G mutation could be responsible for the attenuation seen in Candid#1, despite increased hTfR1 usage. However, while I agree with the authors that other mutations must be contributing to the Candid#1 phenotype, associating this with V64G specifically seems not to be well supported. In particular, this mutation results in “attenuation” (i.e. a growth deficit) when introduced into non-candid#1 JUNV strains (i.e. Romero) but does not do this for Candid#1 itself, which grow just as well as Romero does despite containing this mutation. This suggests that in the native context this mutation is compensated for by yet another layer of mutations. Rather, F427I has been shown to play a leading role in Candid#1 attenuation, with other mutations (e.g. T168A) possibly also contributing. Given our currently very limited knowledge on this issue, it is highly recommended that the authors limit themselves to referring to other changes in Candid#1 that affect its biology and could counterbalance the effect of hTfR1 and leave it at that to avoid overemphasizing specific mutations that in the end may not actually be relevant at all.

Reviewer #3: The authors have appropriately responded to my comments on the original submission.

**Editorial and Data Presentation Modifications?**

Reviewer #1: -

Reviewer #2: My previous comments have been addressed. However, there are a lot of small mistakes in the text, particularly relating to the figure IDs, that need to be carefully fixed. Specific issues that were identified are listed below, but it is also necessary that the authors carefully check their text again for accuracy.

1) Line 134. The phrase ‘utilizes hTfR1 to infect” is duplicated.

2) Table 1. The authors should replace the reference to “fragment S” with “S segment” as this is the standard nomenclature. Please check the manuscript one more time to make sure this is corrected throughout. 

3) Line 976. Shouldn’t this be “L segment” instead of “S segment”?

4) Figure S2A and S4. Shouldn’t this be VSV M that is shown in this Western Blot? Compare text lines 209-211, 982-3, 1009. 

5) Lines 988 and 991-992. These statements regarding the analysis of significance in the data are partly redundant.

6) Lines 211-216. I believe the reference should be to Fig S2 not S1

7) Line 223, 234, 239, 900. Shouldn’t these statements refer to Fig S3 instead of Fig S1/S2?

8) Figure S3. The labelling in panel A is confusing as it contains no reference to being done in 293T cells (unlike all other panels in this figure, which specify the cell line used). 

9) Line 295-8. This statement is not derived from the authors’ results and thus rather belong in the discussion.

10) Line 928. Shouldn’t this be “…colored as in panel B”?

11) Line 360. This should be “293T and A549 cells….(Figs 5B, C, respectively)” since the data for 293T cells are presented first in Fig 5B.

12) Line 940. This should be A549 cells not 293T cells.

13) Line 943. This appears to be the description for panel E, a description of panel D is missing.

14) Line 457. I believe this should be Fig 7B, C.

15) Line 471. A reference to the data in Fig 7D (for the mutants in the TAMV-REF background) is missing here.

16) Line 590. Should this be “mapping analyses (Fig. 4)…”?

17) There appears to be no reference to Text S1 within the manuscript. Also in Text S1, lines 7 and 9, I believe the authors are referring to the data shown in Fig S5 not Fig S2. 

18) Text S2. Lines 8, 10 and 19. I believe the references should be to Fig S6 and not S3.

Reviewer #3: (No Response)

**Summary and General Comments**

Reviewer #1: After addressing the comments of reviewer #2 the manuscript is further improved and I recommend accepting it.

Reviewer #2: It needs to be ensured that the GenBank IDs for TCRV are added in the final version of the manuscript.

Reviewer #3: The authors have appropriately responded to my comments on the original submission.

PLOS authors have the option to publish the peer review history of their article (what does this mean?). If published, this will include your full peer review and any attached files.

Reviewer #1: No

Reviewer #2: No

Reviewer #3: No
---

## [Editor Report · Decision Letter 2]

23 Nov 2020

Dear Dr. Moreno,

We are pleased to inform you that your manuscript 'A NOVEL CIRCULATING TAMIAMI MAMMARENAVIRUS SHOWS POTENTIAL FOR ZOONOTIC SPILLOVER' has been provisionally accepted for publication in PLOS Neglected Tropical Diseases.

Best regards,

Jonas Klingström

Associate Editor

Jeremy Camp

Deputy Editor

---

## [Editor Report · Acceptance letter]

15 Dec 2020

Dear Dr. Moreno,

We are delighted to inform you that your manuscript, "A NOVEL CIRCULATING TAMIAMI MAMMARENAVIRUS SHOWS POTENTIAL FOR ZOONOTIC SPILLOVER," has been formally accepted for publication in PLOS Neglected Tropical Diseases.

Best regards,

Shaden Kamhawi

co-Editor-in-Chief

Paul Brindley

co-Editor-in-Chief
